# Direct comparison of clathrin-mediated endocytosis in budding and fission yeast reveals conserved and evolvable features

Yidi Sun[1], Johannes Schöneberg[1], Xuyan Chen[2], Tommy Jiang[1], Charlotte Kaplan[1], Ke Xu[2], Thomas D Pollard[3,4,5], David G Drubin[1]*

[1]Department of Molecular and Cell Biology, University of California, Berkeley, Berkeley, United States; [2]Department of Chemistry, University of California, Berkeley, Berkeley, United States; [3]Department of Molecular Biophysics and Biochemistry, Yale University, New Haven, United States; [4]Department of Cell Biology, Yale University, New Haven, United States; [5]Department of Molecular Cellular and Developmental Biology, Yale University, New Haven, United States

*For correspondence:
drubin@berkeley.edu

Competing interests: The authors declare that no competing interests exist.

**Abstract** Conserved proteins drive clathrin-mediated endocytosis (CME), which from yeast to humans involves a burst of actin assembly. To gain mechanistic insights into this process, we performed a side-by-side quantitative comparison of CME in two distantly related yeast species. Though endocytic protein abundance in *S. pombe* and *S. cerevisiae* is more similar than previously thought, membrane invagination speed and depth are two-fold greater in fission yeast. In both yeasts, accumulation of ~70 WASp molecules activates the Arp2/3 complex to drive membrane invagination. In contrast to budding yeast, WASp-mediated actin nucleation plays an essential role in fission yeast endocytosis. Genetics and live-cell imaging revealed core CME spatiodynamic similarities between the two yeasts, although the assembly of two zones of actin filaments is specific for fission yeast and not essential for CME. These studies identified conserved CME mechanisms and species-specific adaptations with broad implications that are expected to extend from yeast to humans.

## Introduction

Clathrin-mediated endocytosis (CME) takes up nutrients, regulates responses to extracellular stimuli, and controls the chemical composition and surface area of the plasma membrane. From yeast to humans, cells recruit several dozen highly conserved proteins to CME sites in a regular, predictable order to facilitate plasma membrane internalization (*Engqvist-Goldstein and Drubin, 2003*; *Haucke and Kozlov, 2018*; *Kaksonen and Roux, 2018*; *McMahon and Boucrot, 2011*; *Mettlen et al., 2018*; *Lu et al., 2016*).

During CME, cells deform the plasma membrane locally and subsequently pinch off a vesicle into the cytoplasm. However, the forces required to achieve membrane deformation and the sizes of the resulting vesicles can vary widely between species and cell types (*Lacy et al., 2018*; *McMahon and Boucrot, 2011*). For animal cells, different cell types, and even different areas of the same cell, may exhibit differences in plasma membrane tension (*Boulant et al., 2011*), affecting the forces required to create a membrane invagination. Both budding and fission yeast have high internal turgor pressure, which strongly opposes plasma membrane invagination necessary for endocytosis. Actin filament assembly mediated by the Arp2/3 (actin-related protein 2/3) complex generates force to overcome plasma membrane tension in animal cells (*Boulant et al., 2011*; *Merrifield et al., 2004*; *Merrifield et al., 2002*; *Yarar et al., 2005*) and turgor pressure in yeast (*Aghamohammadzadeh and Ayscough, 2009*; *Basu et al., 2014*; *Kaksonen et al., 2003*;

*Sirotkin et al., 2005*). How various organisms utilize a similar set of proteins to satisfy their particular force generation requirements for endocytosis is an unanswered, fundamental question.

Observations of CME in live cells combined with the superb genetics of yeast, have advanced our understanding to this process. Studies of budding yeast have used electron microscopy (EM) (*Buser and Drubin, 2013*; *Idrissi et al., 2012*; *Kukulski et al., 2012*; *Mulholland et al., 1994*), super resolution microscopy of fixed cells (*Mund et al., 2018*) and comprehensive, high frame rate (*Picco et al., 2015*), live-cell imaging of many of the proteins tagged with fluorescent proteins (*Galletta et al., 2008*; *Kaksonen et al., 2003*; *Kaksonen et al., 2005*; *Newpher et al., 2005*; *Sun et al., 2006*) to produce a detailed protein recruitment time course that was mapped onto precise morphological stages of membrane deformation (such as invagination length and vesicle release) (*Boettner et al., 2012*; *Goode et al., 2015*; *Lu et al., 2016*). Quantitative fluorescence microscopy of live fission yeast also provided counts of endocytic proteins over time (*Sirotkin et al., 2010*; *Arasada and Pollard, 2011*; *Berro and Pollard, 2014*; *Basu and Chang, 2011*). Super resolution imaging of endocytosis in fission yeast provided more details about the process (*Arasada et al., 2018*) and electron microscopy of chemically fixed cells suggested that the endocytic tubules may elongate over 250 nm (*Encinar del Dedo et al., 2014*). While much of what is known about CME was learned from decades of classic cell biology studies in mammalian cells (*Schmid et al., 2014*), more recent advanced experimental approaches, such as genome editing, stem cell technology, and sophisticated new imaging modalities, are beginning to make mammalian cells accessible to studies similar to those performed on yeast (*Cocucci et al., 2014*; *Dambournet et al., 2018*; *Doyon et al., 2011*; *Grassart et al., 2014*; *Kadlecova et al., 2017*; *Liu et al., 2018*; *Sochacki et al., 2017*; *Taylor et al., 2011*).

The ancestors of budding yeast and fission yeast separated about 330–420 million years ago, so these two yeasts are almost as distant from each other as either is to animal cells (*Hedges, 2002*; *Sipiczki, 2000*). Since actin assembly accompanies CME from yeast to humans, this association must be ancient. Given that relating CME protein numbers and dynamics to membrane morphology is still difficult in mammalian cells, a direct, quantitative comparison of CME in the two highly divergent yeasts provides a promising approach to reveal the fundamental and evolutionarily adaptable aspects of the process.

The quantitative data have generated two distinct models for how actin polymerization produces force for endocytosis, a push and pull model for budding yeast (*Sun et al., 2006*) and a two-zone model for fission yeast (*Arasada and Pollard, 2011*) (*Figure 1—figure supplement 1*) (*Lacy et al., 2018*). Precise knowledge of endocytic protein numbers and dynamics is essential for understanding basic mechanisms and how they are adapted by evolution. Therefore, we compared directly the protein numbers and dynamics in the two yeasts using the same methods to provide insights into how the process has been modified during evolution. Quantitation of protein numbers at endocytic sites was a critical, pioneering development in fission yeast studies (*Sirotkin et al., 2010*). Lower numbers of homologous proteins were reported to be present at budding yeast endocytic sites (*Picco et al., 2015*) (*Figure 1—figure supplement 1*), and some differences were reported even for the same protein in the same yeast species (*Table 1*) (*Basu and Chang, 2011*; *Chen and Pollard, 2013*; *Epstein et al., 2018*; *Galletta et al., 2012*; *Picco et al., 2015*; *Sirotkin et al., 2010*), highlighting the importance of direct comparisons. Furthermore, the dynamics of some of the proteins including WASp, one of the main nucleation promoting factors (NPFs) for the Arp2/3 complex, were reported to be distinctly different in budding and fission yeast (*Kaksonen et al., 2003*; *Sirotkin et al., 2010*) (*Figure 1—figure supplement 1*).

Here we compared CME in budding and fission yeast side-by-side by recording the numbers and dynamics of endocytic protein homologs simultaneously at high frame rates by wide-field fluorescence microscopy. A new extracellular standard particle with 120 copies of GFP (*Hsia et al., 2016*) was used to calculate the absolute numbers of some of these proteins at endocytic sites and to compare the relative abundance of multiple endocytic proteins. The live-cell imaging data were processed using a centroid tracking method (*Sbalzarini and Koumoutsakos, 2005*) and then analyzed quantitatively using newly-developed custom software, which detects the different stages of fission yeast endocytic internalization, such as invagination elongation, scission, and vesicle release. Three-dimensional stochastic optical reconstruction microscopy (3D-STORM) of fixed fission yeast cells revealed new structural details of endocytic vesicle formation. Finally, we examined the mechanism

**Table 1.** Comparison of peak protein numbers in actin patches.

Calibration methods: Sirotkin, Arasada, Chen, Epstein, MacQuarrie: a range of internal standards tagged with YFP or GFP; Basu: comparison with actin patches in other cells marked with Arp2/3 complex subunit mYFP-arc5 assumed to be 330 molecules per patch from *Sirotkin et al. (2010)*; Sun: External standard nanoparticle with 120 eGFPs; Galletta: internal Cse4 standard; Picco, Manenschijin: internal Nuf2 standard which was calibrated as a ratio with Cse4.

| Protein | S. pombe | | | | S. cerevisiae | | | |
|---|---|---|---|---|---|---|---|---|
| | Sirotkin et al. (2010)[a]; Arasada and Pollard (2011)[b]; Chen and Pollard (2013)[c]; Epstein et al. (2018)[d] | Basu and Chang (2011) | MacQuarrie et al. (2019) | Sun This study | Sun This study | Picco et al. (2015) Rvs167 125 | Manenschijin et al. (2019) | Galletta et al. (2012) |
| Actin | 7500 [a]; 4050 [b] | | | | | | 3563 | |
| Arp2/3 complex subunits | 320, 320, 320 [a] | ~150 | | 304 (302) | 294 | 210 | 247 | 300 |
| WASp/Las17 | 230 [a]; 140 [b]; 135 [c] | 125 | ~150 | 138 | 102 | 46 | | |
| WIP/Vrp1 | 140 [a] | | | 95 | 78 | | | |
| Myosin-I/ Myo3, Myo5 | 400 [a]; 170 [b]; 240 [c] | | ~350 | 170 | Myo5: 200 Myo3: 100 | Myo5: 132 | Myo5: ~130 Myo3: ~60 | |
| Fimbrin/Sac6 | 910 [a]; 800 [c]; 1000, 1600 [d] | | | 675 | 545 | | 455 | |
| End4/Sla2 (HIP1R) | 160 [a]; 70 [c] | | | 124 | 133 | 37, 47 | ~40 | |
| Pan1/End3 (Intersectin) | Pan1: 260[a]; 160 [c] | | | Pan1: 219 | Pan1: 131 End3: 100 | End3: 60 | | |
| Sla1 | | | | | 168 | 91 | ~90 | |
| ABP1 | | | | | 800 | 423 | 515 | 810 |
| Clathrin HC | 40 [a] | | | | | | | |
| Clathrin LC | 40, 30 [a] | ~40 | | | | | | |
| Capping protein | 230 [a] | | | | | | 197 | 200 |
| App1 | 150 [a] | | | | | | | |
| Coronin | 490 [a]; 320 [b] | | | | | | | |
| Twinfillin | 210 [a] | | | | | | | |
| F-BAR Cdc15 | 125 [b] | | ~70 | | | | | |
| F-BAR Bzz1 | 90 [b] | | | | | | | |
| Dip1 | | 20 | | | | | | |
| Bbc1 | | | ~50 | | | | | |
| Amphiphysin | | | | | | Rvs167 125 | | |

of endocytic actin assembly by performing two-color live-cell imaging on fission yeast mutants with deletions of the Arp2/3 complex-activating CA domains of WASp or type I myosin.

Our quantitative, direct comparison of budding and fission yeast revealed that endocytosis in the two yeasts is remarkably similar in spite of 400 million years of divergent evolution. Differences in the importance of the WASp and type I myosin nucleation promoting factors and the longer invaginations in fission yeast are notable adaptations. These results provide a framework for developing a deeper understanding of CME in these and other species.

## Results

### Side-by-side quantitative ratiometric imaging reveals modest differences in numbers of homologous proteins at endocytic sites in budding and fission yeast

To make accurate comparisons of the numbers of homologous proteins that are recruited to endocytic sites in budding and fission yeast, we recorded the dynamics of endogenously-tagged fluorescent endocytic proteins in both yeasts simultaneously in the same microscope field. We used fission yeast media to culture both yeasts because fission yeast grow poorly in budding yeast media (data not shown), while budding yeast cells grow robustly in fission yeast media and their endocytic proteins with endogenous fluorescent protein tags behave normally (*Figure 1—figure supplement 2*). After determining accurate ratios of these proteins, we used a new calibration standard to convert the measured fluorescence intensities to protein numbers.

The fluorescence intensities of homologous proteins at endocytic sites over time were quantitatively analyzed in both yeasts (*Figure 1—figure supplement 3* for detailed Materials and method). *Figure 1* plots the maximum intensities of seven homologous proteins at endocytic sites in budding and fission yeast including coat proteins and the endocytic actin machinery. Throughout this manuscript we use 'sp' to denote a protein from *Schizosaccharomyces pombe* and 'sc' to denote a protein from *Saccharomyces cerevisiae*. spPan1 and scPan1 are homologues of mammalian intersectins and couple actin assembly to endocytic sites (*Bradford et al., 2015*; *Sun et al., 2015*). spEnd4 and scSla2 are homologs of mammalian HIP1R and couple the endocytic actin network to the plasma membrane and clathrin coat (*Kaksonen et al., 2003*; *Sun et al., 2005*). The maximum amount of spPan1 is approximately 1.7 times that of scPan1, while spEnd4 and scSla2 are present at endocytic sites in similar numbers (*Figure 1A and B*). spWsp1/scLas17 and spVrp1/scVrp1 are yeast homologs of mammalian WASp and WIP, respectively. The maximum levels of spWsp1 and spVrp1 are modestly higher (1.4 and 1.2 fold, respectively) at endocytic sites than scLas17 and scVrp1 (*Figure 1C and D*). scMyo3 and scMyo5 are budding yeast type I myosins. The combined number of scMyo3 and scMyo5 molecules is ~1.8 times the number of spMyo1 molecules at endocytic sites (*Figure 1E*). spArc5 and scArc15 are homologues of mammalian ARPC5, one of the seven subunits of Arp2/3 complex. The numbers of spArc5 and scArc15 molecules recruited to endocytic sites are indistinguishable (*Figure 1F*). The actin filament crosslinking protein spFim1 is ~1.2 fold more abundant in fission yeast than the homologous budding yeast protein, scSac6 (*Figure 1G*).

These measurements show that the numbers of most of the proteins at endocytic sites are ~1.2 to~1.7 fold higher in fission yeast than their homologues in budding yeast (*Figure 1*). The exception is type I myosin, which is present at ~1.8 fold higher levels at budding yeast endocytic sites than at fission yeast sites (*Figure 1E*). Importantly, this side-by-side comparison (*Figure 1* and *Figure 2A* green bar) shows that the differences in numbers is smaller than reported previously (*Picco et al., 2015*; *Sirotkin et al., 2010*) (*Figure 2A*, gray bar). These results motivated us to reexamine the absolute numbers of the endocytic proteins through direct comparison to a common molecular standard.

### Absolute numbers of proteins at endocytic sites quantified using ratiometric comparison to fluorescence intensity of a 120-sfGFP-tagged nanocage

We calibrated our microscope using as a standard particle a hyperstable, water-soluble 60-subunit protein nanocage (~25 nm diameter) with in-frame fusions of sfGFP (super-folder Green Fluorescent Protein (GFP)) to both termini of each subunit, so each particle contains 120 copies of sfGFP (*Hsia et al., 2016*). These particles were expressed in and released from *E. coli* (see Materials and methods for details). The distributions of intensities of the 120-sfGFP-tagged nanocages prepared on different days fit Gaussian distributions with similar means and SDs (*Figure 2—figure supplement 1*). The mean fluorescence intensity was linearly proportional to exposure time over a range from 100 ms to at least 1600 ms per frame (*Figure 2B*). We used 100–500 ms exposure times to image live cells and count molecules. Assuming that the same fraction of sfGFP matures in the bacteria and the two yeast species, the 120-sfGFP-tagged nanocages are expected to enable faithful

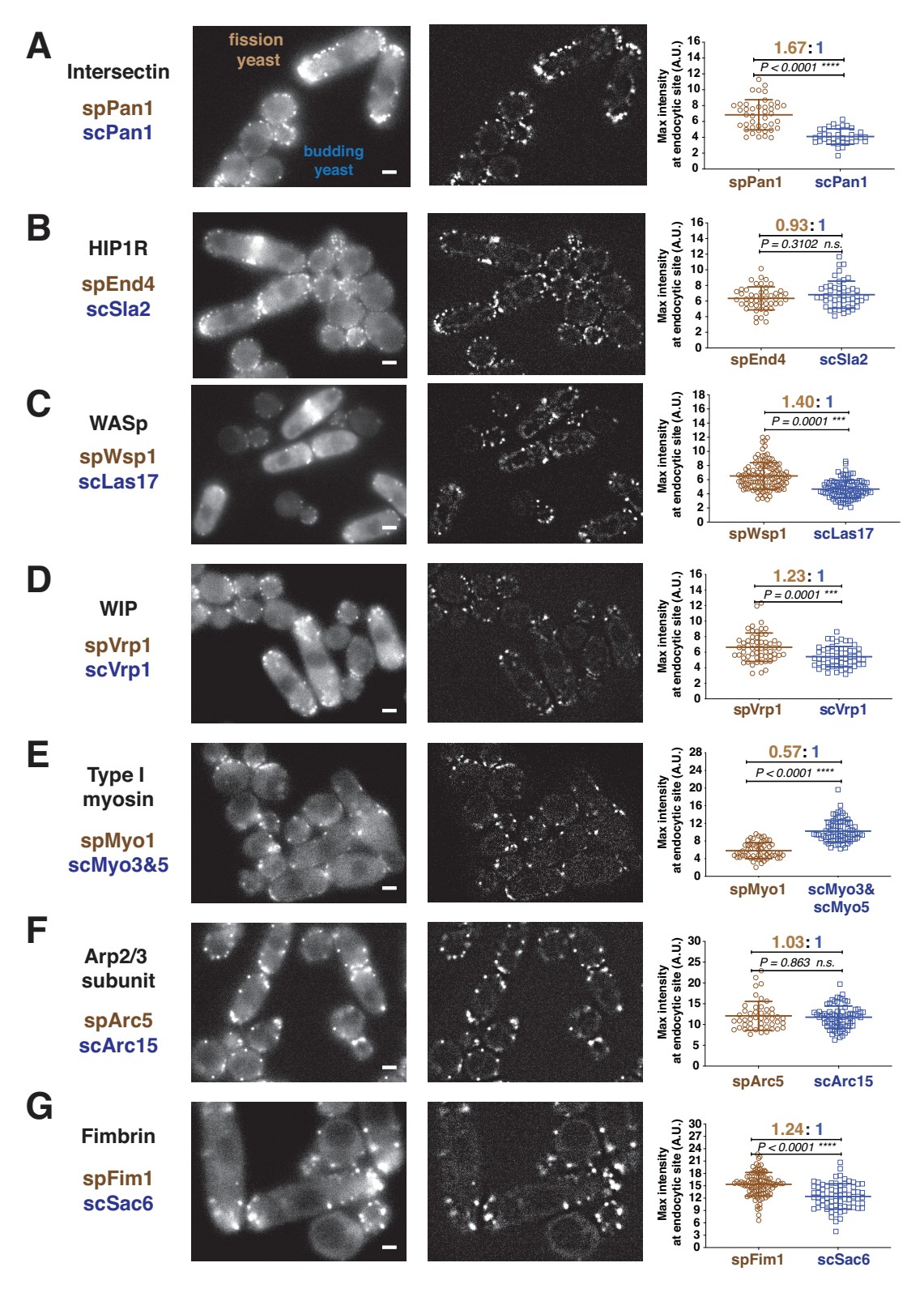

**Figure 1.** Quantitative side-by-side comparison of levels of homologous proteins at endocytic sites in budding and fission yeast. Single frames from unprocessed (left panel) and processed (middle panel) movies for particle intensity quantification, and average maximum intensity of indicated proteins at endocytic sites (right panel). (A-G) Indicated endocytic protein homologues were tagged with GFP or mEGFP in budding yeast or fission yeast, respectively, at the endogenous loci. The maximum intensity for each tagged protein was determined from particle tracking data (see *Figure 1—figure*

*Figure 1 continued on next page*

*Figure 1 continued*

*supplement 3* for details). mEGFP signal is 14% brighter than GFP (*Coffman et al., 2011*). The brightness difference was corrected throughout the data analysis. For most indicated proteins, at least 50 endocytic sites were examined. n.s. stands for 'not significant'. Scale bars on the cell images are 2 μm.

The online version of this article includes the following source data and figure supplement(s) for figure 1:

**Source data 1.** *Figure 1A-G* maximum intensity source data.
**Figure supplement 1.** Endocytic sites in budding and fission yeast: A brief summary of what has been reported previously.
**Figure supplement 2.** Quantitative comparison of fluorescently-tagged scSla1 in budding yeast cultured in different media.
**Figure supplement 3.** Live cell imaging data processing and analysis by Particle Tracker plugin.

determination of numbers of fluorescently-tagged proteins present within several fold of 120 using our imaging setup.

Under identical imaging conditions, the mean maximum fluorescence intensities of the endocytic coat proteins Sla1-sfGFP, Myo5-sfGFP and Las17-sfGFP at numerous endocytic sites in budding yeast cells, and of hundreds of 120-sfGFP-tagged nanocages, were determined (*Figure 2C and D*). Ratiometric comparisons of fluorescence intensity indicate that the mean maximum numbers (± SD) at endocytic sites are 168 ± 48 molecules of Sla1, 199 ± 72 of Myo5 and 102 ± 30 of Las17 (*Figure 2D*).

To validate our use of extracellular 120-sfGFP-tagged nanocages as a standard for measuring intracellular protein numbers, we used the centromere-specific histone H3 variant, Cse4, a well-accepted standard for counting molecules in live cells (*Coffman et al., 2011*; *Lawrimore et al., 2011*). Our measurements indicated that the 16-kinetochore clusters present in living anaphase yeast cells contain 92 ± 19 Cse4-sfGFP molecules (*Figure 2D*), similar to ~80 to~120 molecules counted previously using other calibration methods (*Galletta et al., 2012*; *Lawrimore et al., 2011*).

We next determined maximum numbers of various coat proteins (*Figure 2E*) and proteins of the actin machinery (*Figure 2F and G*) at endocytic sites in both yeasts by comparing the ratios of fluorescence intensities of these proteins with those of Sla1, Las17, or Myo5. We examined more than 20 pairs of fluorescently-tagged proteins side-by-side (*Figure 1*, *Figure 2—figure supplement 2*, *Table 2*, and data not shown), yielding absolute numbers for eight proteins in fission yeast and eleven proteins in budding yeast (*Figures 2E, F and G*, *Table 1*). To evaluate the quality of these measurements, we used both Sla1-GFP and Las17-GFP as standards to determine the number of Myo3-GFPs present at endocytic sites (*Figure 2—figure supplement 3*). The numbers determined using both standards were very similar in two independent experiments (*Figure 2—figure supplement 3*).

The maximum numbers of endocytic proteins we measured for both budding and fission yeast are, to varying degrees, different from, those determined in two previous studies (*Picco et al., 2015*; *Sirotkin et al., 2010*) (*Table 1*). The numbers we determined for budding yeast were generally 1.5–3 times higher than previously reported (*Figure 2E, F and G*) (*Picco et al., 2015*), while the numbers we determined for three key fission yeast proteins (spWsp1, spVrp1 and spMyo1) were 1.5–2 times lower than previously reported by Sirotkin et al. (*Figure 2E, F and G*) (*Sirotkin et al., 2010*). In total, our direct, side-by-side comparison indicates that the numbers of proteins at endocytic sites in these two yeasts are much more similar than was previously thought. The modest differences between the two yeasts may help explain the differences in the dynamics reported in the following sections.

## Particle tracking of fluorescent protein-tagged endocytic proteins reveals different behaviors during internalization in budding and fission yeast

In contrast to the budding yeast, the morphological details of fission yeast endocytic vesicle formation are not well defined. The different stages of fission yeast endocytic internalization (initiation of membrane invagination, membrane elongation and vesicle release) have not been clearly defined. To elucidate such information for fission yeast, we compared the recruitment and spatial dynamics of endocytic proteins side by side in budding and fission yeast. We acquired images in the equatorial plane of cells at high frame rate (more than 7 Hz). The time-lapse images were processed and

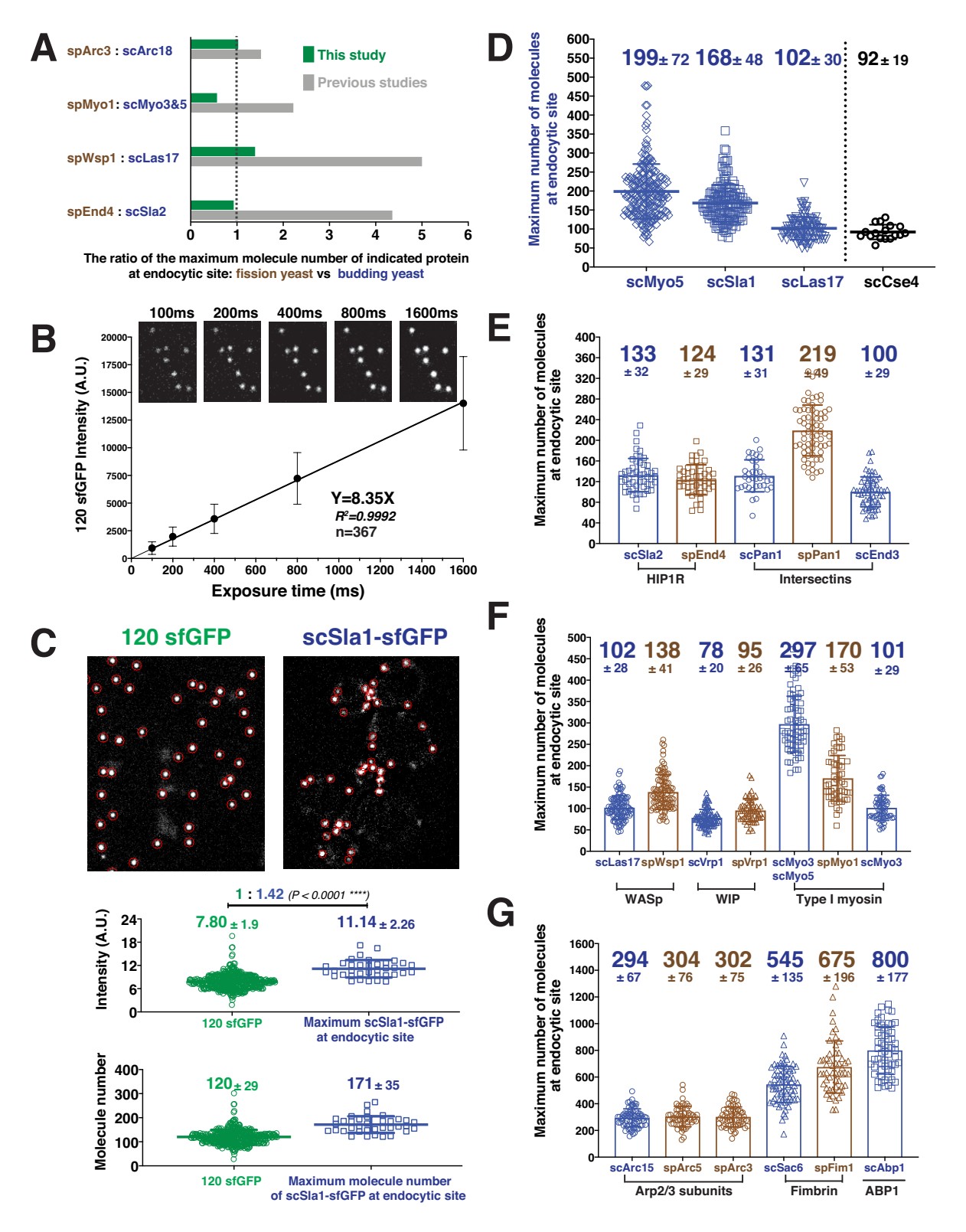

**Figure 2.** Determining the maximum number of fluorescently-tagged endocytic proteins at endocytic sties by ratiometric comparison of fluorescence intensities to the intensity of 120-sfGFP-tagged nanocages. (A) Comparison of ratios of maximum protein levels at endocytic sites for the two yeasts in this study (green) vs in previous studies (gray). (B) The mean fluorescence intensity of 120-sfGFP- tagged nanocages (n = 367) is linearly proportional to exposure time. Images obtained for indicated exposure times are presented at the top of the graph. (C) Ratiometric comparison between the

*Figure 2 continued on next page*

*Figure 2 continued*

fluorescence intensity of 120-sfGFP-tagged nanocages (n = 305) and maximum intensity of scSla1-sfGFP at endocytic sites (n = 58). Left panel, image of 120-sfGFP-tagged nanocages. Right panel, single frame from a processed movie of *scSLA1-sfGFP* cells. The nanocage image and the *scSLA1-sfGFP* movie has been imaged using the same imaging system and the images were processed and analyzed using the same particle tracking parameters. Red circles indicate the sites automatically detected by the tracking program. The intensity (middle) and molecular numbers (bottom) were determined and plotted. (D) Maximum molecular number of scMyo5 (n = 210), scSla1 (n = 152) and scLas17 (n = 112) at endocytic sites as well as the molecular number of scCse4 (n = 17) on the kinetochore clusters. The molecular numbers were calculated by the ratiometric fluorescence intensity comparison of the indicated sfGFP-tagged proteins and 120-sfGFP-tagged nanocages. (E-G) Maximum molecular numbers for indicated proteins at endocytic sites in budding (blue) and fission (brown) yeast. The molecular numbers were calculated by the ratiometric fluorescence intensity comparisons using scSla1, scLas17, or scMyo5 as standards (*Figure 2—figure supplement 2*, *Table 2*). For each indicated protein, at least 50 endocytic sites were examined. The scale bars on the images are 2 µm.

The online version of this article includes the following source data and figure supplement(s) for figure 2:

**Source data 1.** *Figure 2D* maximum protein number source data.
**Figure supplement 1.** Quantitative comparison of 120-sfGFP-tagged nanocages prepared on different days.
**Figure supplement 2.** Quantitative comparison of the maximum number of homologous proteins at endocytic sites in budding and fission yeast.
**Figure supplement 3.** Determining the maximum number of Myo3-GFP molecules using Sla1-GFP and Las17-GFP as standards.

analyzed using the ImageJ Particle Tracker plugin, which records the centroid position and fluorescence intensity of a fluorescently-tagged protein over time to generate a trajectory (*Figure 1—figure supplement 3*). We developed Python-based software to analyze the trajectory data.

The dynamics of the actin filament network were compared in the two yeast species as a function of time by tracking scAbp1-GFP (Actin-Binding Protein 1) in budding yeast and spFim1-mEGFP (fimbrin) in fission yeast in data collected from the same movies at a frame rate of 9 Hz (*Figure 3A*). Both scAbp1-GFP and spFim1-mEGFP patches show little motility at the beginning of their lifetimes in endocytic patches while the fluorescence intensity gradually increases (*Figure 3B and C*). All trajectories for each protein were aligned at 50% of their maximum intensity, then averaged and the average plotted (*Figure 3D and E*). The scAbp1-GFP moves about 125 nm with relatively high regularity (as indicated by the small standard deviations; *Figure 3—figure supplement 1A*) as the intensity approaches the maximum (*Figure 3D*). Once the scAbp1-GFP intensity starts to decline, the movement of the centroid becomes far more irregular (as indicated by a pronounced increase in the standard deviation) (*Figure 3D* and *Figure 3—figure supplement 1A*). Similar results were obtained for scSac6-GFP (data not shown). Previous EM studies suggested that in budding yeast the distance

**Table 2.** The ratio of peak fluorescence intensities between various fluorescently tagged proteins.

| GFP-tagged protein pair | The ratio of peak fluorescence intensities |
|---|---|
| spPan1: scSla1 | 1.30: 1 |
| spPan1: scPan1 | 1.67: 1 |
| scEnd3: spPan1 | 0.46: 1 |
| scSla2: scLas17 | 1.30: 1 |
| spEnd4: scSla2 | 0.93: 1 |
| spWsp: scLas17 | 1.40: 1 |
| scVrp1: scSla1 | 0.47: 1 |
| spVrp1: scVrp1 | 1.23: 1 |
| scMyo3: scMyo5 | 0.51: 1 |
| spMyo1: scMyo3 and scMyo5 | 0.57: 1 |
| spArc5: scSla1 | 1.80: 1 |
| spArc5: scArc15 | 1.03: 1 |
| spArc3: scArc15 | 1.03: 1 |
| scAbp1: scSla1 | 4.80: 1 |
| spFim1: scAbp1 | 0.84: 1 |
| spFim1: scSac6 | 1.24: 1 |

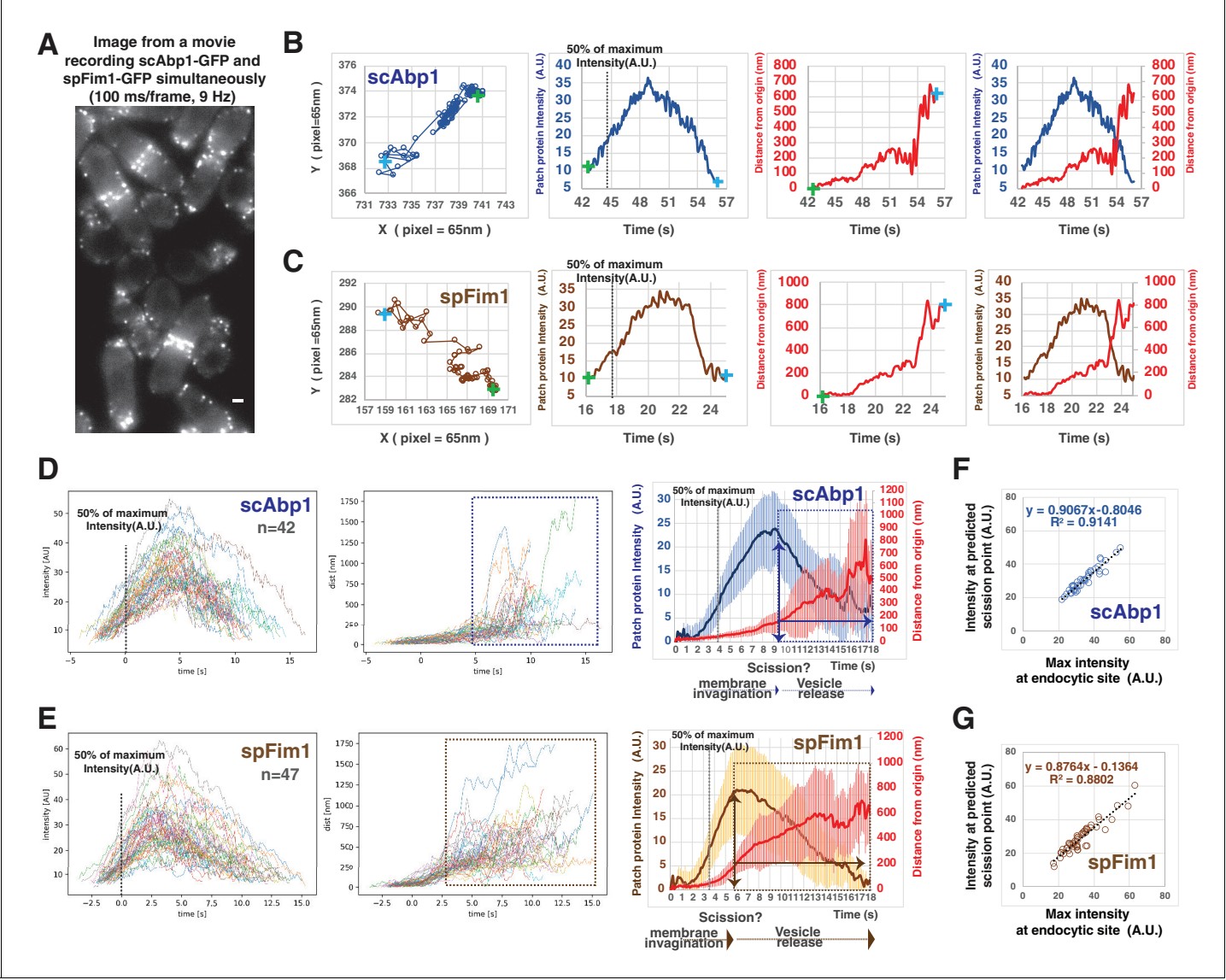

**Figure 3.** Quantitative comparison of endocytic actin dynamics in budding and fission yeast. (**A**) Single frame from a movie of simultaneously imaged *scABP1-GFP* in budding yeast and *spfim1-mEGFP* in fission yeast. (**B and C**) Single endocytic events represented by scAbp1 in budding yeast (**B**) and by spFim1 in fission yeast (**C**) were tracked and then analyzed using our custom software. Graphs from left to right: (Left) Protein patch centroid position over time. Consecutive positions are connected by lines. '+' in green or in blue indicates the first or the last position, respectively. (Left Center) Fluorescence intensity over time. Dotted line indicates the time point when fluorescence intensity reaches 50% of its maximum intensity. (Right Center) Displacement over time. Displacement from the origin is the distance from the position of each time point to the starting position ('+' in green). (Right) Fluorescence intensity and displacement over time. (**D and F**) Numerous endocytic events tracked by imaging fluorescent scAbp1 (**D**) or spFim1 (**E**) were analyzed and aligned. Graphs from left to right: numerous endocytic events aligned to the point of 50% of maximum intensity (indicated by dotted line); displacement data for same cells as in right panel aligned according to 50% maximum intensity point (boxed areas represent inferred movement after scission); combined average results from graphs on the left (note that time and intensity are rescaled on the averaged data graphs). Dotted line indicates the time point when fluorescence intensity reaches to 50%. Vertical line with arrow indicates the inferred moment of scission predicted by the dramatic increase in standard deviation (*Figure 3—figure supplement 1*). Standard deviation is represented by the shadow around the average line. (**F and G**) Inferred endocytic scission is tightly correlated with the time when endocytic actin assembly reaches its maximum for budding (**F**) and fission yeast (**G**). The scale bars on cell pictures are 2 μm.

The online version of this article includes the following figure supplement(s) for figure 3:

**Figure supplement 1.** Predicting the timing of endocytic vesicle scission by the changes of the standard deviation of actin patch displacement.

between the base and the tip of the invaginated endocytic membrane often reaches about ~100 nm in depth before vesicle scission occurs (*Kukulski et al., 2012*). Considering the fluorescence and EM data that reflects on endocytic membrane morphology, we conclude that the transition to irregular movement of scAbp1-GFP beyond ~125 nm from the origin likely represents endocytic vesicle release in budding yeast. Our data for budding yeast agree well with those of *Picco et al. (2015)*, in which Rvs167 (budding yeast amphiphysin) dynamics was used to predict the scission point.

The dynamics of spFim1-mEGFP (*Figure 3E*) followed a similar course as that for scAbp1-GFP (*Figure 3D*). Strikingly, however, the irregular movement of spFim1-mEGFP (indicated by a sharp increase in the standard deviation, *Figure 3—figure supplement 1B*) occurs after its initial regular displacement reaches ~200 nm. Since the irregular movement likely reflects diffusive movement of the free vesicle, endocytic membrane invaginations in fission yeast appear to be substantially deeper than in budding yeast before vesicle scission occurs. Moreover, in both yeasts, the predicted scission time point is tightly correlated with the maximum intensity of the endocytic actin network (*Figure 3F and G*).

Our quantification of actin (represented by Fim1) dynamics predicts the timing of endocytic membrane scission in fission yeast. To determine when endocytic invagination starts, we carried out a similar quantitative analysis for the endocytic coat proteins scSla1-GFP and spPan1-mEGFP, which are closely associated with the invaginating membrane (*Figure 4*). In both yeasts, the coat proteins are stationary for a relatively long period (~30 s –120 s) before moving into the cell interior (*Figure 4A and B*). Presumably, the initiation of coat protein movement reflects the starting point of endocytic membrane invagination. We defined the initiation of coat protein movement by an inflection point in the 'distance from origin' vs 'time' graph (*Figure 4A and B*). Our custom software automatically detects the inflection point for each trajectory and aligns multiple trajectories at the inflection point (*Figure 4C and D*). After the inflection point, scSla1-GFP and spPan1-mEGFP move regularly (indicated by a relatively small standard deviation) for about 4–5 s, followed by a sharp transition to a period of irregular movement (indicated by a pronounced increase in the standard deviation) (*Figure 4C and D*). Thus, we conclude that endocytic vesicle scission likely occurs around 4–5 s after the initiation of endocytic membrane invagination in both yeasts. The scSla1-GFP patches in budding yeast move ~100 nm before vesicle scission (*Figure 4C*). Similar results were obtained for scSla2-GFP, scPan1-GFP and scEnd3-GFP (data not shown). However, in fission yeast, spPan1-mEGFP (*Figure 4D*) and spSla2-mEGFP (data not shown) move ~200 nm during the 4–5 s after the initiation of coat protein movement and before the transition to the irregular movement phase. These results suggest that that the fission yeast endocytic membrane invagination reaches to ~200 nm prior to vesicle scission. While the initiation of both spPan1-mEGFP and scSla1-GFP patch movement is tightly correlated with attainment of the maximum intensity for each protein (*Figure 4E*), the invagination speed in fission yeast is about twice as fast as in budding yeast (51.8 nm/s vs 23.8 nm/s) (*Figure 4F*).

To examine how the onset of the actin assembly relates to the initiation of the endocytic membrane invagination, we performed two-color imaging of spPan1-mEGFP and spFim1-mCherry at a rate of more than seven frames per second in fission yeast (*Figure 5A and B*, *Figure 5—figure supplement 1*). Actin assembly (imaged using spFim1-mCherry) begins when spPan1-mEGFP reaches its peak intensity (*Figure 5B*, dotted line 1). About ~2 s later (*Figure 5B*, dotted line 2), both fluorescent proteins start to move in a regular manner more than 200 nm over 4–5 s prior to beginning to move irregularly when the endocytic vesicle is presumed to be released. At this point the spFim1-mCherry intensity peaks (*Figure 5B*, dotted line3 and *Figure 5—figure supplement 1*). These results indicate that the actin network begins to asssemble slightly before initiation of endocytic membrane invagination, and that assembly peaks around the time when scission occurs. We also performed two-color imaging of spHob1-GFP and spFim1-mCherry at a rate of more than four frames per second in fission yeast (*Figure 5C and D*, *Figure 5—figure supplement 2*). spHob1 is homologue of scRvs167 (amphiphysin) (*Kaksonen et al., 2005*; *MacQuarrie et al., 2019*), which is used as a marker for vesicle scission in budding yeast (*Picco et al., 2015*). Initiation of spHob1-GFP recruitment occurs slightly after initiation of spFim1 recruitment (*Figure 5D*, dotted line 1) and rapidly reaches its peak intensity (*Figure 5D*, dotted line 2). spHob1-GFP remains at a high intensity level until spFim1 reaches its peak intensity (*Figure 5D*, dotted line 3). The dynamics of spHob1 are consistent with a role in vesicle scission, analogous the role of scRvs167 (*Kaksonen et al., 2005*).

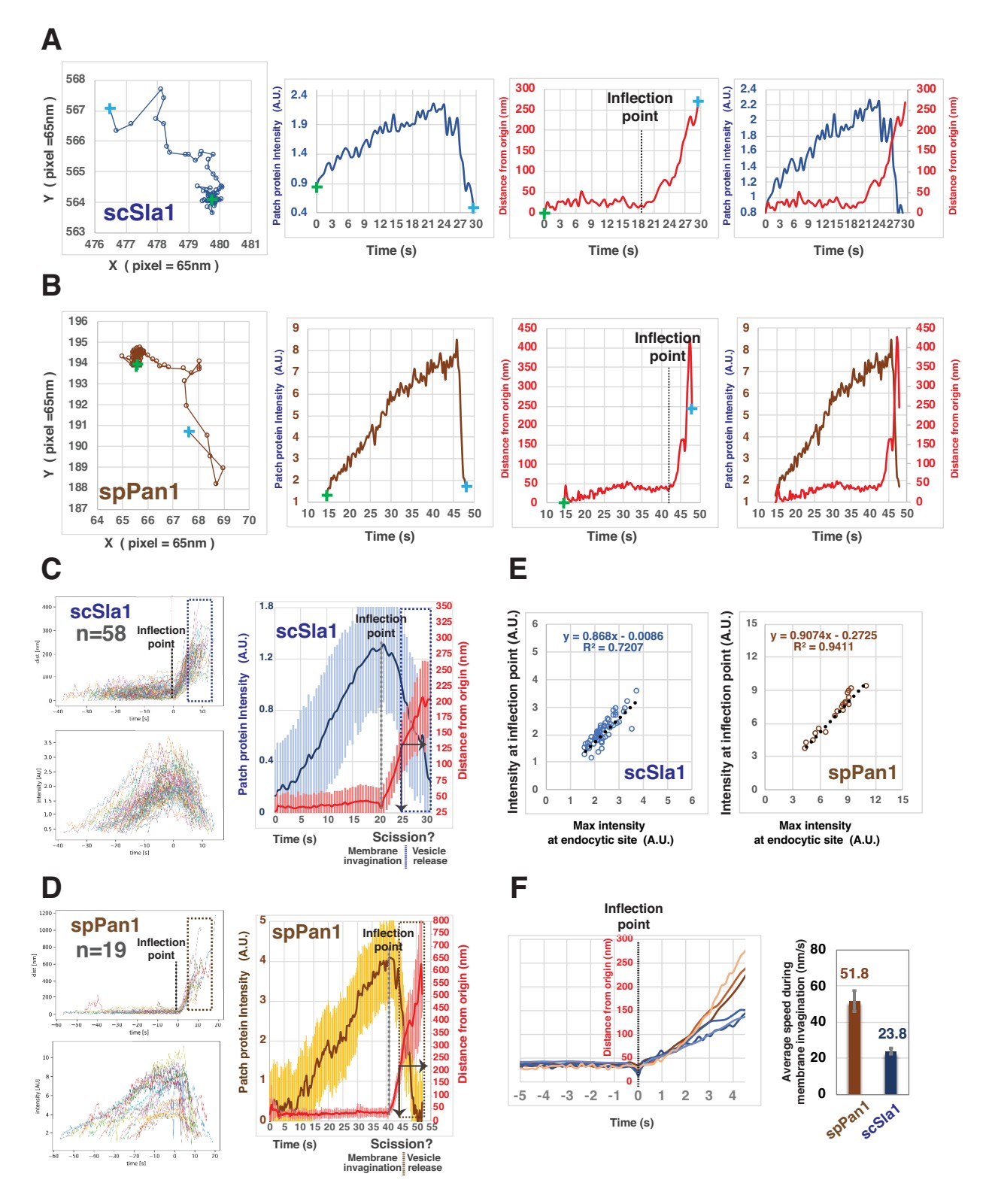

**Figure 4.** Quantitative comparison of endocytic coat dynamics for budding and fission yeast. (**A and B**) Single endocytic event detected by tracking fluorescent scSla1 in budding yeast (**A**) or spPan1 in fission yeast (**B**). Graphs from left to right: protein patch centroid position over time, '+' in green or in blue indicate the first and last positions, respectively; fluorescence intensity over time; displacement over time ('+' in green and blue indicate starting and end points, respectively while the dotted line indicates the inflection point at which time the coat protein begins to move; fluorescence intensity

*Figure 4 continued on next page*

*Figure 4 continued*

and displacement over time. (**C and D**) Numerous endocytic events detected by tracking fluorescent scSla1 (**C**) or by spPan1 (**D**) were analyzed and aligned. Graphs from left to right: numerous endocytic events aligned at inflection points indicated by the dotted line, the boxed area represents movement after inferred scission event; fluorescence intensity aligned by the movement inflection point; combined average results from the two graphs on the left (note that time and intensity are rescaled in the average graph and that the dotted line indicates inflection point), vertical line with arrow indicates the moment of scission predicted by the dramatic standard deviation increase. Standard deviation is represented by the shadow around the average line. (**E**) Initiation of endocytic membrane invagination is tightly correlated with the time when the endocytic coat reaches its maximum amount in budding and fission yeast. (**F**) Speed of coat protein movement during the inferred invagination process prior to scission. Each of the brownish or bluish lines represents the average displacement of spPan1 or scSla1, respectively, over time from one experiment. Three independent experiments were performed for the indicated proteins. Bar graphs show rates for coat proteins.

Together, our quantitative analysis clearly defined the timing of different stages during endocytic internalization in fission yeast and revealed intriguing differences in endocytic invagination development between budding and fission yeast. The results demonstrate that both yeast spend similar time deforming the membrane prior to scission. However, the resulting endocytic invagination of fission yeast appears to be nearly twice as long as in budding yeast.

## Three-dimensional stochastic optical reconstruction microscopy (3D-STORM) reveals novel nanoscale structural details of endocytic vesicle formation in fission yeast

To acquire a nanoscale structural description of endocytic site formation in fixed fission yeast cells, we carried out 3D-STORM analysis by labeling GFP-tagged endocytic proteins with AF647-conjugated anti-GFP nanobodies in fixed cells (*Kaplan and Ewers, 2015*; *Mund et al., 2014*). Nanobodies conjugated to the AF647 are small enough to readily penetrate the yeast cell wall and thus eliminate the requirement for its prior enzymatic removal, allowing high quality structural preservation and reduced localization error.

Numerous endocytic sites labeled by spFim1-mEGFP or spPan1-mEGFP in fixed cells were examined at the equatorial plane of the cells by 3D-STORM (*Figure 6A–6D*, *Figure 6—figure supplement 1*), so invaginations of the plasma membrane were in the image plane. spFim1 was used to label the endocytic actin network. The length of the spFim1 structures ranged from 120 nm to 350 nm (*Figure 6A*), which likely represents progression through the endocytic pathway (*Arasada et al., 2018*). In a similar STORM study in fixed budding yeast, the length of the endocytic actin network was estimated to range from 70 nm-240 nm (*Mund et al., 2018*). EM studies in budding yeast suggest that a dense network of actin filaments occupies a large area outside the invaginated membrane prior to scission. This area is expected to be larger than that traveled by the centroid of actin network fluorescence in the dynamics experiments described above because those studies do not describe the extent of the network but instead its center. Thus, the greater length of endocytic actin structures observed in fission yeast is consistent with the possibility that the invaginated membrane that it covers is deeper than in budding yeast. The length of spPan1 structures in fission yeast ranged from 50 nm to 180 nm (*Figure 6B*), which is much smaller than the range for spFim1 structures. spPan1 is a coat protein that is expected to localize on the invaginated membrane, more precisely representing the morphology of invaginated membrane. In addition, 3D-STORM analysis revealed that spPan1 forms a ring structure in the X-Y plane, presumably encircling the invaginated membrane (*Figure 6C*). A similar ring structure could also be observed with the spFim1-mEGFP labeling (*Figure 6—figure supplement 1D*). Moreover, in the equatorial plane we observed structures that we interpret as pinched-off endocytic vesicles based on their distances of >500 nm from the cell cortex (*Figure 6D*). The average longest axis of endocytic vesicles was 193 ± 37 nm when labeled by spFim1 and 152 ± 30 nm for spPan1. A similar nanoscale structural analysis is currently not available for budding yeast endocytic vesicles.

Together, our 3D-STORM results support the conclusion that fission yeast endocytic invaginations are deeper than those observed in budding yeast.

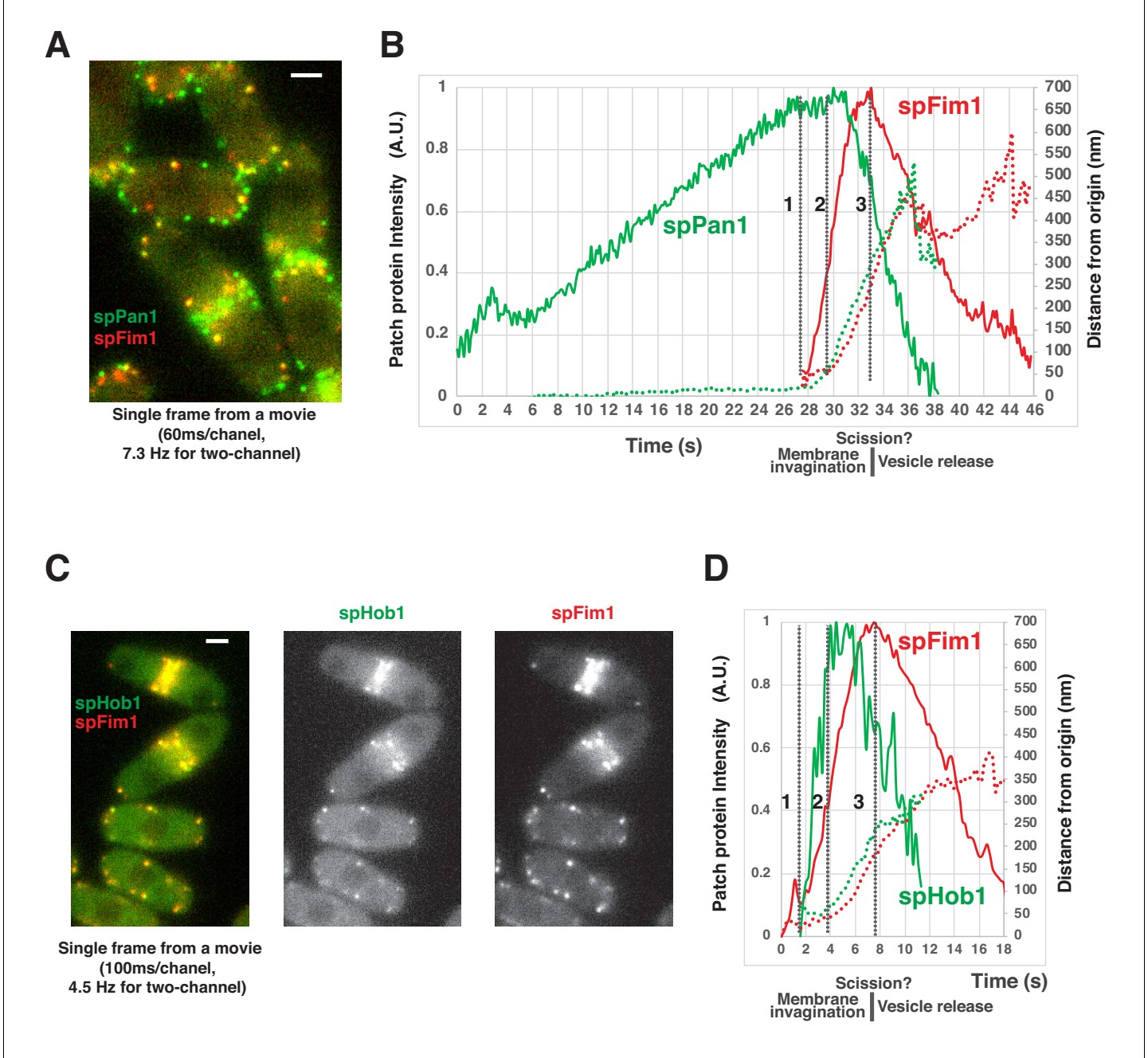

**Figure 5.** Spatial-temporal relationship of coat proteins, scission proteins and the actin network at fission yeast endocytic sites. (**A**) Single frame from a movie of spPan1-mEGFP and spFim1-mCherry expressed in fission yeast cells. (**B**) Alignment of average intensity (solid lines) and displacement (dotted lines) of spPan1-mEGFP (coat) and spFim1-mCherry (actin cytoskeleton) patches (n = 15) (*Figure 5—figure supplement 1* for details). Vertical dotted line one indicates the time when actin assembly is first detected. Vertical dotted line two marks the inferred initiation of membrane invagination. Vertical dotted line three indicates the inferred scission event. (**C**) Single frame from a movie of spHob1-GFP spFim1-mCherry expressed in fission yeast cells. (**D**) Alignment of average intensity (solid lines) and displacement (dotted lines) of spHob1-GFP (scission protein) and spFim1-mCherry (actin cytoskeleton) patches (n = 23) (*Figure 5—figure supplement 2* for details). Vertical dotted line one indicates the time when spHob1-GFP is first detected. Vertical dotted line two marks inferred initiation of membrane invagination. Vertical dotted line three indicates inferred scission event. Scale bars on cell pictures are 2 µm.

The online version of this article includes the following figure supplement(s) for figure 5:

**Figure supplement 1.** Alignment and quantification of average trajectories for spFim1-mCherry and spPan1-mEGFP in fission yeast.
**Figure supplement 2.** Alignment and quantification of average trajectories for spFim1-mCherry and spHob1-GFP in fission yeast.

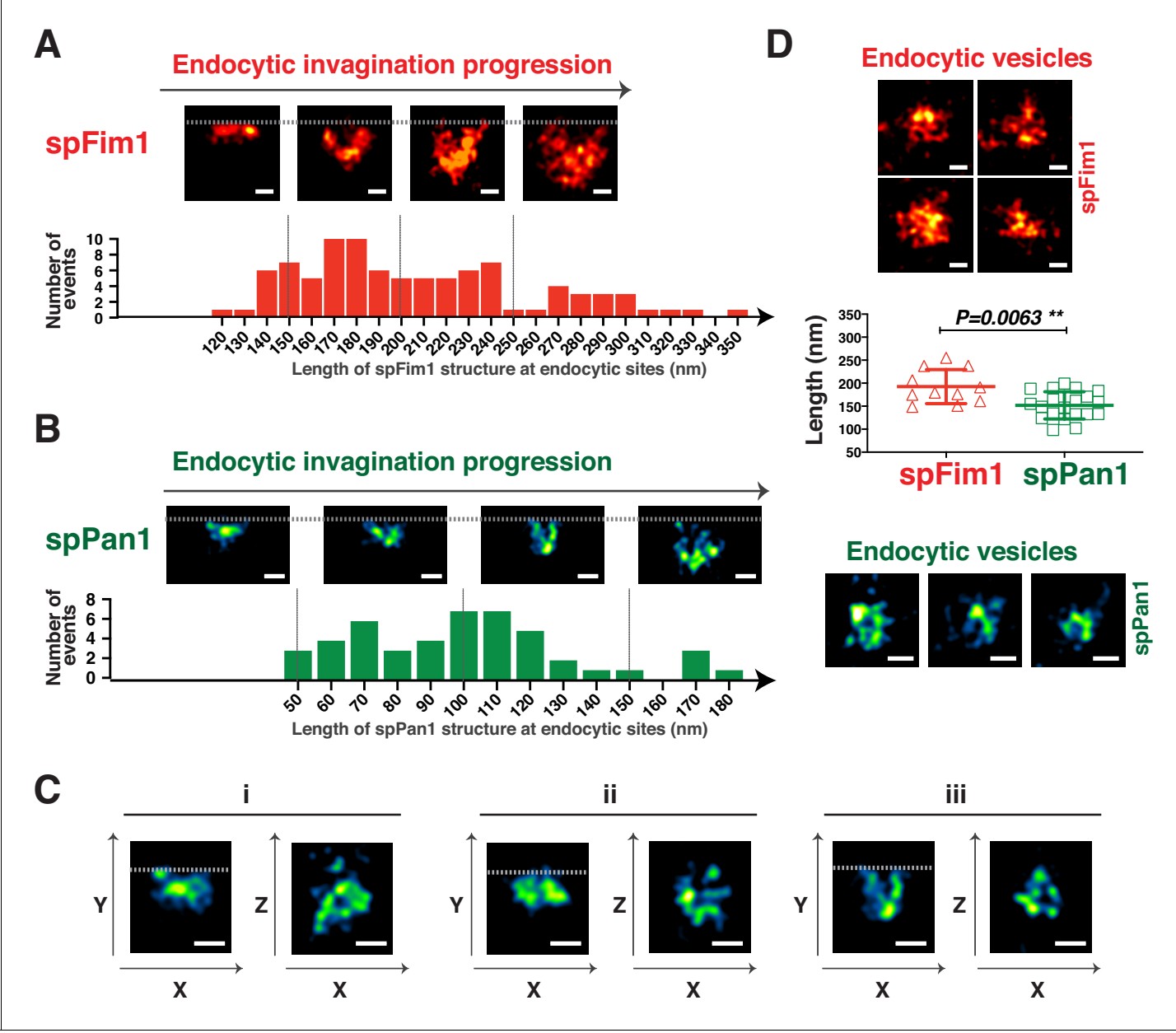

**Figure 6.** 3D-STORM imaging of coat protein and the actin network at fission yeast endocytic sites. (**A and B**) The length of nanoscale structures of spFim1 (n = 92) or spPan1 (n = 46) at endocytic sites revealed by STORM analysis was measured and plotted. Representative STORM images of different length and quantitative analysis of spFim1 (**A**) and spPan1 (**B**). The histogram shows the frequency distribution of observed endocytic structure lengths. C, 3D-STORM image analysis reveals ring-like Pan1 organization in the XZ dimensions. D, Representative images of presumed endocytic vesicles in the cytoplasm. The length of longest axis of each presumed vesicle was measured and plotted. Fim1-labeled structures were pseudo-colored red hot (**A and D**) and Pan1-labeled structures were pseudo-colored green fire blue (**B, C, D**). The scale bars on STORM images are 100 nm. The online version of this article includes the following figure supplement(s) for figure 6:

**Figure supplement 1.** 3D-STORM imaging of spFim1 and spPan1.

## Spatial-temporal relationship between two major Nucleation Promoting Factors (NPFs) and the actin network during fission yeast endocytosis

We next compared the actin assembly mechanisms that drive endocytic internalization in the two yeasts. We focused on WASp (spWsp1) and type I myosin (spMyo1), which in fission yeast are the two major NPFs that activate the Arp2/3 complex at endocytic sites (*Sirotkin et al., 2005*). Budding

yeast WASp, scLas17, in has been reported to remain near the base of the endocytic pit (close to the plasma membrane) as the actin network moves into the cytoplasm (*Kaksonen et al., 2003*), while fission yeast WASp, spWsp1, was reported to move away from the base with the actin network (*Sirotkin et al., 2010*). One possible source of the different observations on these two yeasts is that a fluorescent protein-tag was added at C-terminus of budding yeast scLas17 but to the N-terminus of fission yeast spWsp1. Therefore, we generated a budding yeast strain with the GFP tag on the N-terminus of Las17 (*Figure 7—figure supplement 1*) by making an in-frame fusion at the native *LAS17* locus. More than 70% of GFP-Las17 patches showed dynamics similar to those observed for Las17-GFP (*Figure 7—figure supplement 1A and B*). However, some GFP-Las17 patches appeared to split into two parts toward the end of the patch lifetime: the majority of the GFP-Las17 signal remained non-motile until it disappeared from the endocytic site while a small amount of the GFP-Las17 moved away from the cell surface (*Figure 7—figure supplement 1A*). In spite of these differences, the populations of actin patches in the *GFP-LAS17* strain and the *LAS17-GFP* strain were indistinguishable by two criteria: (1) patch lifetimes were identical; and (2) the actin network markers Abp1p-RFP and Myo5p-RFP behaved similarly over space and time in the two strains (*Figure 7—figure supplement 1C*) (*Sun et al., 2006*).

Our high frame rate live-cell image analysis confirmed that mEGFP-spWsp1 patches move away from their origin, into the cytoplasm in fission yeast (*Figure 7A*). Intriguingly, our newly developed alignment software revealed that mEGFP-spWsp1 only begins to move when it accumulates to its maximum level (*Figure 7A*). mEGFP-spWsp1 patches then move ~200 nm over ~4 s, while their fluorescence intensity rapidly declines to the baseline level (*Figure 7A*). The regular phase of spWsp1 movement prior to the onset of the irregular movement corresponds well with the endocytic invagination dynamics observed with other fission yeast proteins (*Figure 3*, *Figure 4*, and *Figure 5*). These observations (*Figure 7—figure supplement 1* and *Figure 7A*) confirm that the differences between WASp dynamics in the two yeasts are real rather than resulting from differences in the fluorescent fusion proteins or differences in the analytical methods.

To better understand how the two NPFs regulate endocytic actin assembly, we investigated the spatiotemporal relationship between the spFim1 and spWsp1 or spMyo1 using high time resolution (7.3 Hz) two-color imaging (*Figure 7B–7E*, *Figure 7—figure supplement 2*, *Figure 7—figure supplement 3*). spFim1 dynamics appear to be identical in *spfim1-mEGFP*, *mEGFP-spwsp1 spfim1-mCherry*, and *mEGFP-spmyo1 spfim1-mCherry* strains (*Figure 3G*, *Figure 7C and E*), indicating that mEGFP fusions do not affect spWsp1's or spMyo1's function or actin patch dynamics. mEGFP-spWsp1 appears at the cell cortex first, then actin assembles (spFim1-mCherry) only when mEGFP-spWsp1 accumulates to approximately 50% of the maximum level (*Figure 7C*, dotted line 1). Once spWsp1 reaches the maximum level, both spWsp1 and the actin network marker spFim1 begin to move (*Figure 7C*, dotted line 2). While spWsp1 and the actin network move ~200 nm from the origin, the spWsp1 signal declines to the baseline level while the actin network approaches the maximum level (*Figure 7C*, dotted line 3). Consistently, vesicle scission appears to occur when actin assembly reaches its maximum level (*Figure 3G*, *Figure 5B and D*, *Figure 7C*, and *Figure 7—figure supplement 2*). spMyo1 accumulates at endocytic sites with similar timing to spWsp1, using spFim1 accumulation as a temporal alignment reference (*Figure 7D and E* and *Figure 7—figure supplement 3*). However, as in budding yeast (*Sun et al., 2006*), spMyo1 stays non-motile during its lifetime while the actin network moves into the cytoplasm (*Figure 7D and E*).

We conclude that actin assembly is initiated when both spWsp1 and spMyo1 reach approximately 50% of their peak levels in fission yeast. Importantly, spWsp1 and spMyo1 are recruited to their maximum levels before the membrane detectably begins to invaginate, supporting the conclusion that spWsp1 and spMyo1 both promote actin nucleation to initiate membrane deformation. Moreover, the rapid disassembly of spWsp1 and spMyo1 from endocytic sites supports the notion that actin nucleation by the two NPFs rapidly decreases while the invagination elongates.

## The relative importance of spWsp1 and spMyo1 NPF activity for fission yeast endocytosis

The previously proposed two-zone model for fission yeast implies that spWsp1 and spMyo1 define two independent pathways for actin assembly at the endocytic sites to facilitate endocytic invagination (*Arasada and Pollard, 2011*) (*Figure 1—figure supplement 1*). Deletion of either Myo1 or Wsp1 stops endocytosis (*Arasada and Pollard, 2011*) but removes not only their NPF activity but

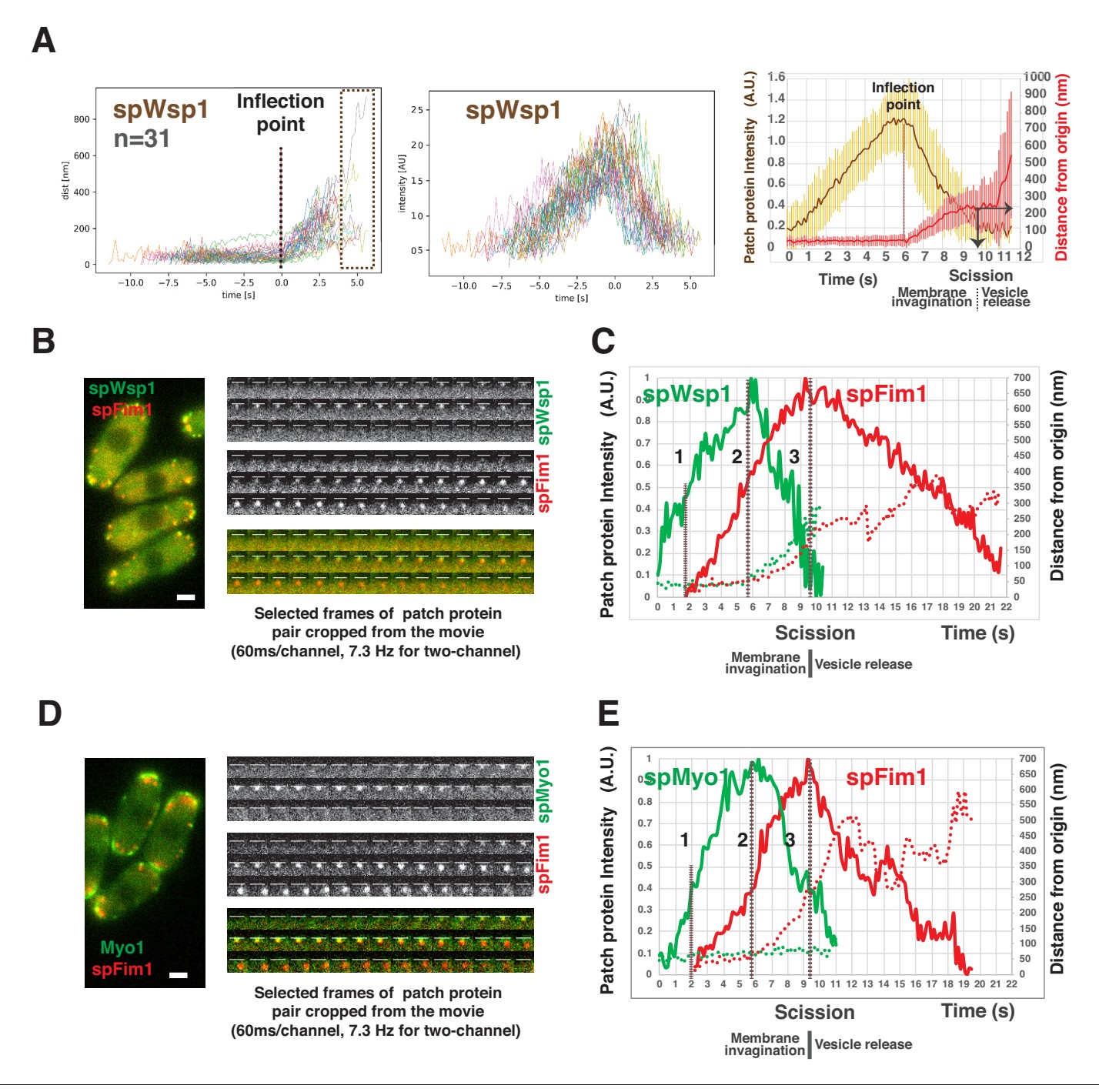

**Figure 7.** Spatio-temporal relationship between two major NPFs and endocytic actin assembly in fission yeast. (A) Dynamics of spWsp1. Numerous endocytic events represented by mEGFP-spWsp1 were analyzed and aligned. Graphs from left to right: (right) Protein position vs time for endocytic events (n = 31) aligned at inflection points indicated by the dotted line. The boxed area represents movement after presumed scission event. (middle) Fluorescence intensity aligned on basis of alignment in right graph. (left) Averaged results from data in two graphs on the left. Note that time and intensity are rescaled in this graph. The dotted line indicates inflection point. Vertical solid line with arrow indicates the moment of scission predicted by the dramatic standard deviation increase. (B) Single frame from a movie of fission yeast expressing mEGFP-spWsp1 spFim1-mCherry (left panel). Time series showing composition of a single endocytic site (right panels). (C) Alignment of average intensity and displacement for mEGFP-spWsp1 and spFim1-mCherry patches (n = 8) (*Figure 7—figure supplement 2* for details). Dotted line one indicates actin assembly initiation. Dotted line two indicates initiation of membrane invagination. Dotted line three indicates inferred scission. (D) Single frame from a movie of fission yeast expressing mEGFP-Myo1 and spFim1-mCherry (left panel). Time series showing composition of a single endocytic site (right panels). (E) Alignment of average

*Figure 7 continued on next page*

*Figure 7 continued*

intensity and displacement for mEGFP-spMyo1 and spFim1-mCherry patches (n = 9) (*Figure 7—figure supplement 3* for details). Dotted line one indicates actin assembly initiation. Dotted line two indicates initiation of membrane invagination. Dotted line three indicates inferred scission. Scale bars are 2 µm.

The online version of this article includes the following figure supplement(s) for figure 7:

**Figure supplement 1.** Dynamics of GFP-scLas17 and scLas17-GFP in budding yeast.
**Figure supplement 2.** Alignment and quantification for average trajectories for spFim1-mCherry and mEGFP-spWsp1 in fission yeast.
**Figure supplement 3.** Alignment and quantification for average trajectories for spFim1-mCherry and mEGFP-spMyo1- in fission yeast.

also other functions (*Pedersen and Drubin, 2019*; *Sun et al., 2006*) provided by domains that do not interact with the Arp2/3 complex. To specifically test the importance of the NPF activity, we created mutants of Myo1 and Wsp1 only lacking the carboxy terminal 'CA' motifs that bind and activate Arp2/3 complex.

The spWsp1CAΔ-GFP mutant protein accumulates and persists at endocytic sites nearly three times longer than the wild-type protein without the pronounced movement detectable in the wild type cells (*Figure 7A*, *Figure 8A and B*). Two-color live cell imaging demonstrated that the actin network (spFim1-mCherry) still assembles, but that 92% (142 out of 154) of patches with assembled actin are non-motile during their lifetimes, indicating that endocytic internalization is strongly compromised in the *spwsp1-CAΔ* mutant (*Figure 8C and D*, *Video 1*, and *Video 2*). In contrast, endocytic internalization still occurs in the *spmyo1CAΔ* mutant (*Figure 8E and F*, *Video 3*, and *Video 4*), in which its 'CA' motif was precisely truncated at its genomic locus. Consistently, a recent study showed that the spMyo1 CA domain is not essential for normal coat protein (spEnd4) dynamics (*MacQuarrie et al., 2019*). Moreover, the lifetimes of Myo1CAΔ-GFP and wild-type mEGFP-spMyo1 in patches are indistinguishable (*Figure 8B*).

Together, these observations indicate that spWsp1 NPF activity is essential for endocytosis, that spMyo1 NPF activity is not, and they suggest that two-zone actin assembly is not required for endocytosis in fission yeast.

## Discussion

This study quantitatively compared several important aspects of CME in distantly related yeasts to gain evolutionary insights into underlying mechanisms and how they can be adapted for the specific demands of different organisms and cell types including cells of multicellular organisms.

### Quantitation of endocytic protein numbers

Knowing how many of each type of protein is present at an endocytic site is crucial for developing a quantitative understanding of CME. Previous studies reported larger numbers (up to five fold) of homologous endocytic proteins in fission yeast than in budding yeast (*Picco et al., 2015*; *Sirotkin et al., 2010*) (*Table 1*). However, these counting studies were done in several different laboratories, which focused exclusively on either budding or fission yeast, and they largely used different methods for calibration. In fact, different results have been reported for the number of molecules of the same protein in the same yeast species (*Table 1*) (*Basu and Chang, 2011*; *Chen and Pollard, 2013*; *Epstein et al., 2018*; *Galletta et al., 2012*; *Picco et al., 2015*; *Sirotkin et al., 2010*). To reconcile these differences, we made great efforts to validate our methods for an accurate quantitative analysis of protein numbers, and to make direct comparisons between the yeasts.

Firstly, we determined the relative abundance of homologous proteins in the two yeasts in the same microscope fields to eliminate imaging inconsistencies. The fluorescence intensities were processed and analyzed in parallel, further ensuring a faithful comparison. For these reasons, we believe that our approach improves the accuracy of comparisons between species. Strikingly, the abundances of the endocytic protein homologues in the two yeasts are more similar than indicated by the published literature (*Figure 1*, *Figure 2A*, *Table 1*, and *Table 2*).

Secondly, we adopted 120-sfGFP-tagged nanocages (*Hsia et al., 2016*) as our fluorescent standard for counting absolute protein numbers. Each nanocage is a stable icosahedron with a diameter of ~25 nm. Like endocytic patches, the nanocages appear as diffraction limited spots by 2D-widefield fluorescence microscopy. Nanocages are easy to purify and handle and as a result, large sample

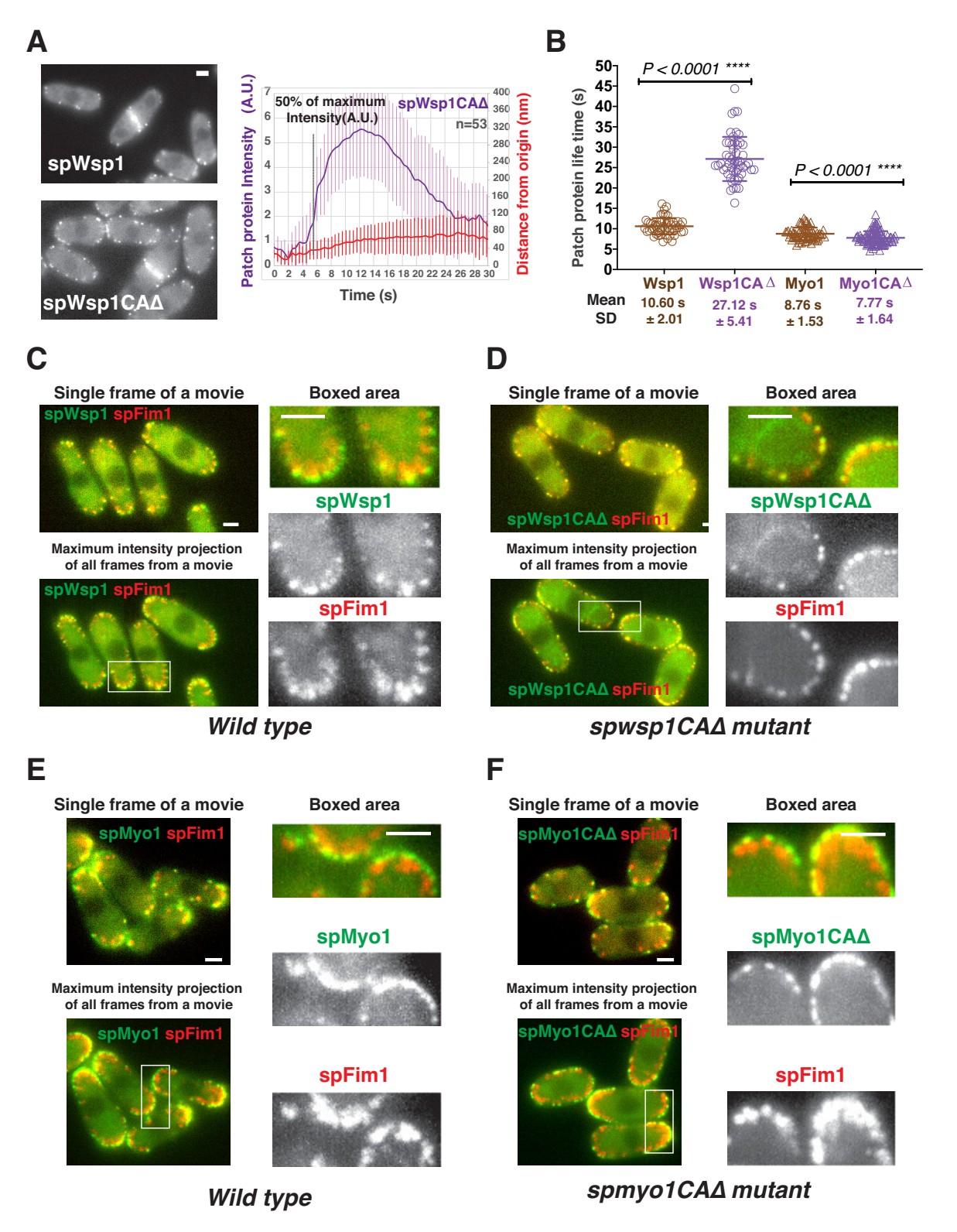

**Figure 8.** Functional analysis of CA domains in two fission yeast nucleation promoting factors. (**A**) Dynamics of spWsp1CAΔ-GFP patches. Single frame from movie of cells expressing mEGFP-spWsp1 (left upper panel) or spWsp1-CAΔ-GFP (left bottom panel). Alignment of average intensity and displacement for spWsp1-CAΔ-GFP patches (n = 53) (right panel). Dotted line indicates 50% maximum fluorescence intensity. (**B**) Patch protein lifetimes for indicated proteins. (**C and D**) Endocytic internalization is severely impaired in *spwsp1-CAΔ* mutants. A single frame (upper left) and Maximum

*Figure 8 continued on next page*

**Figure 8 continued**

intensity projection of all frames (lower left) from a movie of cells expressing mEGFP-spWsp1 spFim1-mCherry (**C**) or spWsp1-CAΔ-GFP spFim1-mCherry (**D**). The maximum intensity projections reveal extent of patch protein movement over time. Enlarged views of the boxed-areas shown in left panels are shown on right. E and F, Endocytic internalization appears unaffected by the *spmyo1-CAΔ* mutation. A single frame (upper left) and maximum intensity projection of all frames (lower left) from a movie of cells expressing mEGFP-spMyo1 spFim1-mCherry (**E**) or spMyo1-CAΔ-GFP spFim1-mCherry (**F**). The maximum intensity projections reveal the extent of patch protein movement over time. Enlarged views of the boxed-areas shown in left panels are shown on the right. Scale bars are 2 μm.

sizes can be imaged in the same field, giving more reliable and precise results providing that the medium has a pH of ~7.0 (details in Materials and methods). Moreover, intensity histograms for 120-sfGFP-tagged nanocages prepared on different days give highly reproducible results (*Figure 2—figure supplement 1*). As validation for our new standard, we used the 120-sfGFP-tagged nanocages to estimate that clusters of 16-kinetochores in budding yeast carry 92 ± 19 Cse4-sfGFP molecules, consistent with results of several previous studies (*Coffman et al., 2011*; *Galletta et al., 2012*; *Lawrimore et al., 2011*).

Our counts of Abp1 and the Arp2/3 complex in actin patches of budding yeast agree well with those of *Galletta et al. (2012)* (*Table 1*), who used the centromere protein Cse4 as an internal standard. We also mostly agree with *Picco et al. (2015)* and *Manenschijn et al. (2019)* on the relative abundances of seven budding yeast endocytic proteins but our counts are mostly ~2 fold higher than theirs (*Table 1*, *Table 2*). These systematic differences in absolute number estimates are likely due to use of different calibration standards. Picco et al. used the kinetochore protein Nuf2 at the anaphase to telophase transition as a standard. However, a recent study demonstrated that Nuf2 levels change dynamically during the anaphase to telophase transition (*Dhatchinamoorthy et al., 2017*), which complicates its use as a standard. For fission yeast, our counts of Fim1, spWsp1 and spMyo1 are substantially lower than some (*Epstein et al., 2018*; *Sirotkin et al., 2010*) but not all (*Arasada and Pollard, 2011*; *Chen and Pollard, 2013*) previous measurements (*Table 1*). These differences might be explained by the live cell imaging conditions and methods to correct for local background.

To date, few counts of proteins have been reported at sites of CME in animal cells due to technical difficulties associated with genetic manipulation of genomes. However, dynamin numbers have been determined in genome-edited mammalian cells (*Cocucci et al., 2014*; *Grassart et al., 2014*). Pertinent to the current study, Akamatsu et al (under revision) recently generated a cell line that expresses an endogenously-tagged fluorescent subunit of Arp2/3 complex and estimated its peak number at endocytic sites as ~200, which is lower than, but close to, the peak numbers at CME sites in budding (294) and fission (302) yeast (*Figure 2G*). Thus, the numbers of endocytic proteins at CME sites in yeast may approximate the numbers in human cells, and differences may provide a basis for understanding cell type-specific requirements for CME such as the forces necessary for vesicle formation.

## Comparison of protein dynamics and endocytic site morphology for CME internalization stages

We analyzed fission yeast endocytic internalization using high time resolution live-cell imaging and custom analytical software. Wide-field fluorescence microscopy in the equatorial plane of yeast cells enabled fast acquisition, high signal intensity and the opportunity to observe the complete endocytic internalization process from a side view. However, the endocytic protein imaged by this method may not remain in the same focal plane during the full-time course, especially after endocytic vesicle scission. For this reason and because of variation in kinetic profiles of different events, the fluorescence

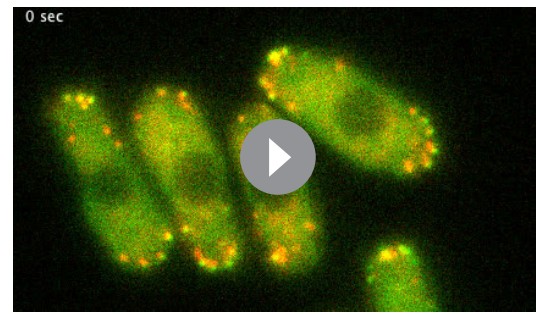

**Video 1.** Dynamics of mEGFP-spWsp1 and spFim1-mCherry in *mEGFP-spwsp1 spfim1-mCherry S. pombe* cells. Time to acquire one image pair was 136 ms. Interval between frames is 548 ms.
https://elifesciences.org/articles/50749#video1

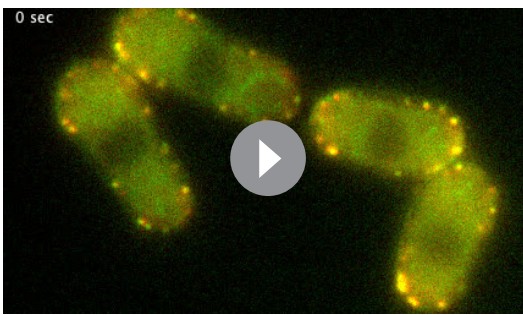

**Video 2.** Dynamics of spwsp1CAΔ-GFP and spFim1-mCherry in *spwsp1-CAΔ-GFP spfim1-mCherry S. pombe* cells. Time to acquire one image pair was 400 ms. Interval between frames is 460 ms.
https://elifesciences.org/articles/50749#video2

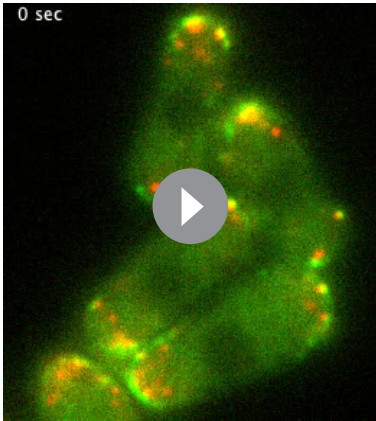

**Video 3.** Dynamics of mEGFP-spMyo1 and spFim1-mCherry in *mEGFP-myo1 spfim1-mCherry S. pombe* cells. Time to acquire one image pair was 200 ms. Interval between frames is 222 ms.
https://elifesciences.org/articles/50749#video3

intensity profiles varied from one recorded trajectory to another. We therefore developed a conceptually different alignment method from those used previously (*Berro and Pollard, 2014*; *Picco et al., 2015*) to suit our imaging method. Our custom software automatically transformed the tracking data for numerous individual events into visually intuitive graphs, showing the 2-D trajectory, raw (unprocessed) fluorescence intensity, and displacement over time. Comparing graphs for numerous events allowed us to identify the signatures for each endocytic protein and align the trajectories of the proteins accordingly. Our software not only recapitulates the dynamic endocytic protein profiles reported previously (*Berro and Pollard, 2014*; *Picco et al., 2015*), but also distinguishes the stages of endocytic internalization, such as invagination elongation, scission, and vesicle release.

In the early studies of fission yeast, the trajectories of most endocytic protein(s) were aligned relative to the initiation of patch movement in movies of lower time resolution (0.3–0.5 Hz) (*Arasada and Pollard, 2011*; *Sirotkin et al., 2010*) without knowing how this movement correlates with specific endocytic steps. A later study improved the temporal resolution and provided an in-depth analysis of Fim1 dynamics (*Berro and Pollard, 2014*). However, in the absence of a comparative analysis of actin and coat protein spatiodynamics, knowledge of how the actin (Fim1) timeline maps onto the morphological stages of CME was still unclear in fission yeast. Our high time resolution movies (more than 7 Hz) and new alignment software revealed that the centroid position of the endocytic actin network (represented by spFim1), the endocytic coat (represented by spPan1) and the presumed scission protein (spHob1) (*Kaksonen et al., 2005*) move ~200 nm from their initiation sites prior to predicted vesicle release (*Figure 3*, *Figure 4*, and *Figure 5*). These results suggest that the endocytic membrane invagination is ~200 nm deep at the time of scission, nearly twice the depth observed for budding yeast. The nanoscale structural organization of spFim1 and spPan1 at endocytic sites determined by super-resolution imaging further supports this interpretation (*Figure 6A and B*). This possibility agrees with the observation of membrane invaginations over 200 nm long in the chemically fixed fission yeast cells (*Encinar del Dedo et al., 2014*). In

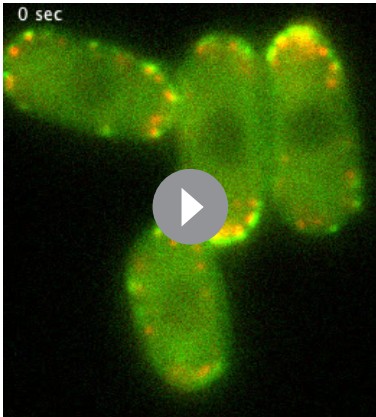

**Video 4.** Dynamics of spMyo1CAΔ-GFP and spFim1-mCherry in *spmyo1CAΔ-GFP spfim1-mCherry S. pombe* cells. Time to acquire one image pair was 200 ms. Interval between frames is 222 ms.
https://elifesciences.org/articles/50749#video4

addition, the initiation of membrane invagination is tightly correlated with the maximum intensity for spPan1-mEGFP, while the predicted scission time point is tightly correlated with the maximum intensity of Fim1 (*Figure 3E and G*, *Figure 4D and E*, *Figure 5A and B*). Our results agree with and provide an explanation for the previous finding that the motion of Fim1 patches is diffusive rather than directional after the peak number at endocytic sites is reached (*Berro and Pollard, 2014*). Another intriguing result is that the elongation speed for membrane invagination is twice as fast in fission yeast as in budding yeast (*Figure 4F*).

The two yeasts studied here show dramatic differences in endocytic site dimensions and dynamics. As similar differences have been reported within a range of mammalian cell types (*Dambournet et al., 2018*; *McMahon and Boucrot, 2011*), comparative studies between the yeasts might identify mechanisms for controlling CME structure and dynamics more generally. Compared to budding yeast, the greater abundance of key proteins at endocytic sites (*Table 1*, *Table 3*) in fission yeast, might account for faster invagination rates and greater invagination lengths. In the future, direct comparisons of turgor pressure between the yeasts and determination of how cargo size and concentration affect rates and vesicle capacity are likely to provide additional insights into the mechanistic basis for observed differences.

## 'push and pull' vs 'two-zone' models for force production by actin polymerization

A two-zone model, in which distinct zones of actin filament nucleation mediated by type I myosin and WASp generate two separate actin networks that push against each other, was proposed to generate forces for endocytosis in fission yeast (*Arasada and Pollard, 2011*) (*Figure 1—figure*

**Table 3.** Peak molecules per patch vs total molecules per cell for various endocytic proteins in budding and fission yeast.

**A. Peak molecules per patch vs total molecules per cell in budding yeast**

| *S. cerevisiae* Protein | Total molecules per cell | Peak molecules per patch | Ratio |
|---|---|---|---|
| | *Ho et al. (2018)* | Sun (This study) | |
| Arp2/3 complex subunits | 12,784, | 294 | 0.023 |
| WASp/Las17 | 3134 | 102 | 0.033 |
| WIP/Vrp1 | 3559 | 78 | 0.021 |
| Myosin-I/ Myo3 and Myo5 | Myo5, 6051 Myo3, 4222 | Myo5:200 Myo3:100 | 0.033 0.024 |
| Fimbrin/Sac6 | 24,543 | 545 | 0.022 |
| HIP1R/Sla2 | 8879 | 133 | 0.015 |
| Intersectin /Pan1 and End3 | Pan1:9830 End3:6064 | Pan1: 131 End3:100 | 0.013 0.016 |
| Sla1 | 8776 | 91 | 0.010 |
| Abp1 | 18,301 | 800 | 0.044 |

**B. Peak molecules per patch vs total molecules per cell in fission yeast**

| *S. pombe* Protein | Total molecules per cell | Peak molecules per patch | Ratio |
|---|---|---|---|
| | *Sirotkin et al., 2010*; *Wu and Pollard, 2005* | Sun (This study) | |
| Arp2/3 complex subunits | ~40,000 | 304 (302) | 0.008 |
| WASp/Wsp1 | 68,000 | 138 | 0.002 |
| WIP/Vrp1 | 19,000 | 95 | 0.005 |
| Myosin-I | 63,000 | 170 | 0.003 |
| Fimbrin/Fim1 | 87,000 | 675 | 0.008 |
| HIP1R/End4 | 22,000 | 124 | 0.006 |
| Intersectin/Pan1 | 27,000 | 219 | 0.008 |

supplement 1). However, when we eliminated the actin filament nucleation promoting activity of spMyo1, spWsp1 alone was sufficient to support robust endocytic internalization with only one zone of actin assembly (*Figure 8F*; *Figure 1—figure supplement 1*). In addition, we found that membrane deformation starts when the numbers of spWsp1 and spMyo1 peak at endocytic sites (*Figure 7*) supporting the conclusion that spWsp and spMyo1 promote actin nucleation to initiate membrane deformation. Interestingly, both yeasts accumulate similar maximum numbers of Arp2/3 complexes (~300) and HIP1R homologs (scSla2 and spEnd4,~130) at endocytic sites (*Figure 2E and F*), suggesting similar capacities for producing actin filament branches and for connecting the actin network to the endocytic coat.

While in mammalian cells the detailed dynamics of N-WASP and type I myosin during CME membrane deformation are not clear, both localize at mammalian CME sites (*Krendel et al., 2007*; *Merrifield et al., 2004*). However, mammalian type I myosin does not contain a CA motif, and so it is unlikely to activate Arp2/3 complex directly and two zones of actin assembly may not be required for mammalian CME.

## Conclusions from quantitative, evolutionary comparison of CME in distantly related yeasts

Integrating our quantitative comparative analysis of endocytosis with previous knowledge revealed several key similarities and differences in budding yeast and fission yeast in terms of molecular numbers, architecture and function of the endocytic coat and associated actin machinery (*Figure 9*).

The first conclusion is that the two yeasts trigger actin assembly at endocytic sites similarly. The coat proteins spPan1, spEnd4, scSla1, scPan1, scEnd3 and scSla2 reach their maximum numbers at endocytic sites before actin assembly is initiated (*Figures 4C, D and E*, *Figure 9A (I and II), 9B (I and II)* and data not shown). Recruitment of spWsp1 at endocytic sites in fission yeast begins as spPan1 peaks (*Figure 5B* and *Figure 7C*, *Figure 9A (I)*) followed in both yeasts by switch-like activation of the Arp2/3 complex when ~ 70 molecules of WASp are recruited to endocytic sites (*Figure 7B and C*, *Figure 9A(II) and B(II)*) (*Sun et al., 2017*). These properties suggest that ~70 WASp molecules are key for establishing a WASp-Arp2/3 complex stoichiometry sufficient for a burst of actin assembly (*Case et al., 2019*).

Second, the two yeasts recruit type I myosin and WASp to endocytic sites differently. The assembly/disassembly profiles of mEGFP-spWsp1 and mEGFP-spMyo1 are similar in fission yeast (*Figure 7B–7E*, *Figure 9A (I)*), while the two budding yeast type one myosins, scMyo3 and scMyo5, appear at endocytic sites more than 10 s after scLas17 (*Figure 9B (I)*) (*Sun et al., 2006*).

Third, WASp-mediated actin assembly initiates membrane deformation in both yeasts. Endocytic membrane invagination is first detected when spWsp1 peaks and actin assembles to ~50% of its maximum (*Figure 7B and C*, and *Figure 9A (III), 9B(III)*).

Fourth, although both yeasts recruit WASp and myosin-I to endocytic sites, the yeast show different dependencies on the NPF activities of these two proteins. Budding yeast recruit a total of 300 molecules of two isoforms of myosin-I (vs. 170 in fission yeast) and the NPF activity of their CA motifs is important for robust internalization (*Sun et al., 2006*) but not in fission yeast (*Figure 8B and F*). Nevertheless, complete deletion of fission yeast spMyo1 shows that, as in budding yeast, functions other than its NPF activity contribute to endocytosis (*Arasada and Pollard, 2011*; *Pedersen and Drubin, 2019*; *Sun et al., 2006*). On the other hand, endocytic sites in fission yeast accumulate more WASp molecules (~140 Wsp1) than in budding yeast (~100 Las17) (*Figure 2F*), and the WASp NPF activity is crucial for endocytosis in fission yeast (*Figure 8A, B and D*) but not budding yeast (*Sun et al., 2006*).

Fifth, once membrane invagination begins, both WASp and myosin-I dissociate more quickly from actin patches in fission yeast (*Figure 7C*, *Figure 7E* and *Figure 9A (IV) and 9A(V)*) than in budding yeast (*Figure 9B(IV) and 9B(V)*). However, the actin network peaks upon vesicle scission in both yeasts.

Finally, the two yeasts elongate the endocytic membrane invagination over the same time before scission, but the invagination grows two-fold faster and twice as long in fission yeast.

Our studies bring together two decades of research on endocytosis in yeast to illustrate how 300 million years of divergent evolution has resulted in two systems using essentially the same complex set of conserved proteins somewhat differently to achieve a common result, formation of an endocytic vesicle. Further work will reveal the specific properties and advantages of each design solution

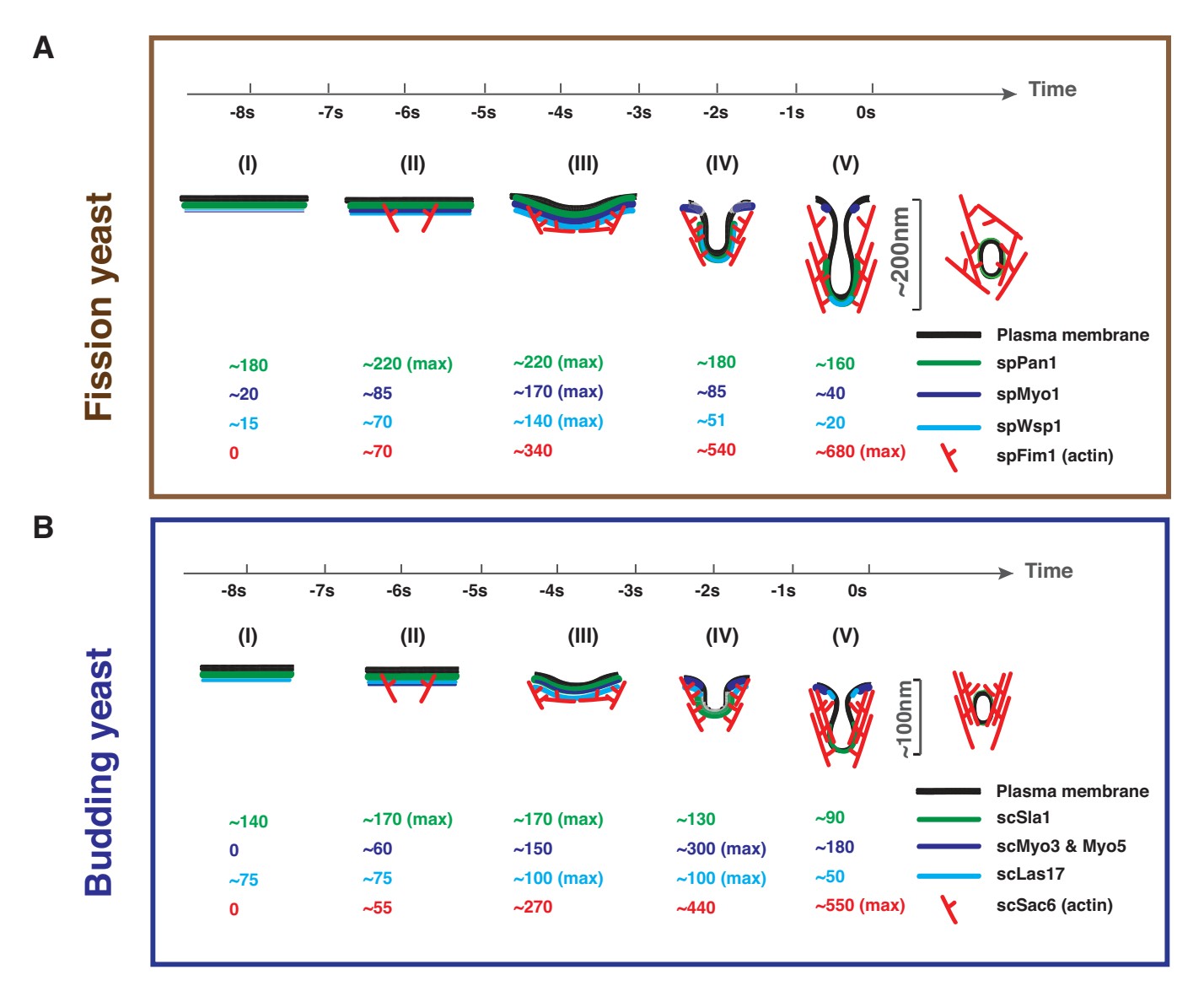

**Figure 9.** Comparison of endocytic vesicle formation in fission and budding yeast. Timeline and summary of the average molecule numbers for indicated coat proteins and actin machinery components in fission (**A**) and budding yeast (**B**) endocytosis. Scission occurs at the time 0. Steps (I) through (V) represent the full process from membrane invagination initiation to vesicle scission. See text (last section in Discussion) for detailed key similarities and differences in fission yeast and budding yeast in terms of protein numbers, architecture and function of the endocytic coat and associated actin machinery.

and will compare the designs of these two systems with those in other organisms and different cell types in multicellular organisms.

## Materials and methods

### Media and strains

*S. cerevisiae* and *S. pombe* strains were both grown in fission yeast standard rich media (YES 225, Sunrise Science Product) or synthetic media (EMM, MP Biomedicals). All strains used in this study are listed in *Table 4*. To generate an *S. cerevisiae* strain expressing N-terminal GFP-tagged Las17, the *GFP* sequence followed by a gctgccgcagcggct linker was chromosomally inserted before the

**Table 4.** Strains used in this study.

| | |
|---|---|
| DDY2733* | *MATa his3-Δ200 leu2-3, 112 lys2-801 ura3-52 ABP1-GFP::HIS3* |
| DDY2734* | *MATa his3-Δ200 leu2-3, 112 lys2-801 ura3-52 SLA1-GFP::HIS3* |
| DDY2735* | *MATα his3-Δ200 leu2-3, 112 lys2-801 ura3-52 PAN1-GFP::HIS3* |
| DDY2795* | *MATa his3-Δ200 leu2-3, 112 lys2-801 ura3-52 SLA2-GFP::HIS3* |
| DDY2736* | *MATa his3-Δ200 leu2-3, 112 lys2-801 ura3-52 LAS17-GFP::HIS3* |
| DDY3201* | *MATa his3-Δ200 leu2-3, 112 lys2-801 ura3-52 VRP1-GFP::HIS3* |
| DDY2752* | *MATα his3-Δ200 leu2-3, 112 lys2-801 ura3-52 ARC15-GFP::HIS3* |
| DDY2960* | *MATα his3-Δ200 leu2-3, 112 lys2-801 ura3-52 SAC6-GFP::HIS3* |
| YSY4303 | *MATα his3-Δ200 leu2-3, 112 lys2-801 ura3-52 LAS17-sfGFP::HIS3* |
| YSY4305 | *MATα his3-Δ200 leu2-3, 112 lys2-801 ura3-52 MYO5-sfGFP::HIS3* |
| YSY4307 | *MATα his3-Δ200 leu2-3, 112 lys2-801 ura3-52 SLA1-sfGFP::HIS3* |
| YSY4315 | *MATα his3-Δ200 leu2-3, 112 lys2-801 ura3-52 CSE4-sfGFP::HIS3* |
| YSY4218 | *MATα his3-Δ200 leu2-3, 112 lys2-801 ura3-52 MYO3-GFP::HIS3* |
| YSY4299 | *MAT a his3-Δ200 leu2-3, 112 lys2-801 ura3-52 MYO3-GFP::HIS, ABP1-RFP::HygMX* |
| YSY4300 | *MAT α his3-Δ200 leu2-3, 112 lys2-801 ura3-52 MYO5-GFP::HIS, ABP1-RFP::HygMX* |
| YSY4125 | *MAT a his3-Δ200 leu2-3, 112 lys2-801 ura3-52 GFP-LAS17::KanMX6, ABP1-RFP::HygMX* |
| YSY4126 | *MAT α his3-Δ200 leu2-3, 112 lys2-801 ura3-52 GFP-LAS17::KanMX6, MYO5-RFP::HIS* |
| YSY4291 | *MAT α his3-Δ200 leu2-3, 112 lys2-801 ura3-52 Myo5-GFP::KanMX6, MYO3-GFP::HIS* |
| YSY4120 | *MAT α his3-Δ200 leu2-3, 112 lys2-801 ura3-52 GFP-LAS17::KanMX6* |
| YSY4257 | *h+ leu1-32 ura4-D18 his3-D1 ade6-M210 pan1-mEGFP-kanMX6 fim1mcherry-natRMX6* |
| YSY4266 | *h- or h+ leu1-32 ura4-D18 his3-D1 ade6 kanMX6- Pwsp1-mEGFP-wsp1 fim1mcherry-natRMX6* |
| YSY4269 | *h- or h+ leu1-32 ura4-D18 his3-D1 ade6 kanMX6- Pmyo1-mEGFP-myo1 fim1mcherry-natRMX6* |
| YSY4335 | *h+ leu1-32 ura4-D18 his3-D1 ade6-M210 fim1mcherry-natRMX6 wsp1-CAΔ-GFP-kanMX6* |
| YSY4345 | *h- or h+ leu1-32 ura4-D18 his3-D1 ade6-M210 myo1-caΔ-GFP-kanMX6 fim1-mcherry-natRMX6* |
| YSY4346 | *h- leu1-32 ura4-D18 his3-D1 ade6-M210 fim1mcherry-natRMX6, hob1-GFP-kanMX6* |
| YSY4346 | *h+ leu1-32 ura4-D18 his3-D1 ade6-M210 hob1-GFP-kanMX6* |
| TP401** | *h+ leu1-32 ura4-D18 his3-D1 ade6-M210 pan1-mEGFP-kanMX6* |
| TP398** | *h+ leu1-32 ura4-D18 his3-D1 ade6-M210 end4-mEGFP-kanMX6* |
| TP203** | *h- leu1-32 ura4-D18 his3-D1 ade6-M210 kanMX6- Pwsp1-mEGFP-wsp1* |
| TP186** | *h- or h+ leu1-32 ura4-D18 his3-D1 ade6-M210 vrp1-EGFP-kanMX6* |
| TP198** | *h- leu1-32 ura4-D18 his3-D1 ade6-M210 kanMX6-myo1-mEGFP* |
| TP226** | *h- leu1-32 ura4-D18 his3-D1 ade6-M210 arpc5-mEGFP-kanMX6* |
| TP347** | *h- leu1-32 ura4-D18 ade6-M210 fim1-mEGFP-kanMX* |
| * | from *Kaksonen et al. (2003)* or *Sun et al. (2006)* |
| ** | from *Sirotkin et al. (2005)* or *Wu and Pollard (2005)* |
| otherwise | *This study* |

*LAS17*-coding region by Gibson assembly cloning (NEB Inc) and yeast transformation. The resulting strain was verified by DNA sequencing.

## Preparation of 120-sfGFP-tagged nanocages from *E. coli*

10 O.D.$_{600}$ (optical density) *E. coli* expressing sfGFP-tagged nanocages were collected and washed twice with, and then diluted into, 500 ul EMM media containing 50 mM Tris (pH 7.4) and one protease inhibitor tablet (Roche). Sonication (1 s on and 1 s off for 1 min, then on ice for 1 min, repeating for 3 times) was used to break cells. The sonicated *E. coli* were centrifuged for 5 min at 14,000 rpm at 4°C and the supernatant was then collected. The supernatant was further diluted 10 times into the

EMM media; we found that maintaining a pH in the range of 7.0 is important for this standard to accurately reflect intracellular molecule numbers. 5 ul diluted supernatant was transferred onto a coverslip and the nanocages were allowed to settle for 10 min prior to imaging.

## Fluorescence microscopy

Fluorescence microscopy was performed using a Nikon Eclipse Ti microscope (Nikon Instruments, Melville, NY) controlled by Metamorph software (Molecular Devices, Sunnyvale, CA) and equipped with a Plan Apo VC 100×/1.4 Oil OFN25 DIC N2 objective (with Type NF immersion oil, Nikon), a Perfect Focus System (Nikon), and a Neo sCMOS camera (Andor Technology Ltd., South Windsor, CT) (65 nm effective pixel size). For live cell imaging, cells were grown overnight in YES 225 rich media. The saturated overnight cell culture was diluted into EMM synthetic media and cells were grown to early log phase at 25°C. When *S. pombe* and *S. cerevisiae* strains were imaged simultaneously, equal numbers of the two yeasts were mixed together just before imaging. The cells were transferred to a 1% agarose pad made from EMM synthetic media. All imaging was done at the equatorial plane of the cells at room temperature. For single-channel live-cell imaging, images were acquired continuously at 2–9 frames/s. Two-channel movies were made using the SPECTRA X Light Engine (Lumencor, Beaverton, OR) for excitation with a 524/628 nm dual-band bandpass filter for GFP/mCherry emission (Brightline, Semrock, Lake Forest, IL). Time to acquire one image pair was 137 ms, or 222 ms.

For the experiment shown in the upper panel of *Figure 1—figure supplement 2A*, the budding yeast cells were grown to early log phase at 25°C in synthetic budding yeast media. The cells were adhered to the surface of a concanavalin A-coated coverslip, which was then inverted onto a glass slide, and sealed with vacuum grease (Dow Corning).

## Image analysis

Image J software was used for general processing of images and movies, such as background subtraction and photobleaching correction (*Kaksonen et al., 2003*). A median filter was used to subtract local background around patches as described in *Picco and Kaksonen (2017)*. Particle Tracker plugin was used to track endocytic patch protein dynamics (*Sbalzarini and Koumoutsakos, 2005*) (*Figure 1—figure supplement 3*). The trajectory selection was confirmed by visual inspection. Trajectories that any point in their lifetime were too close to another patch to be clearly resolved were excluded from our analysis. The sample size for each experiment is shown on the figure or/and in the figure legend. The sample sizes used in this study are more than what have been used in other landmark papers in this type of study (*Kaksonen et al., 2003*; *Picco et al., 2015*; *Sirotkin et al., 2010*). For each pair of variables, pooled data of analysis were compared by a two-sided Mann-Whitney test using the Prism 8 graphing software.

We built a custom data analysis pipeline that operates on individual tracks listed in the trajectory selection. For each track, the intensity values ($I_t$), and the positions in x ($x_t$) and y ($y_t$) for each time point (t) of the track are recorded. Different CME proteins might show different dynamics during their intensity time course and, depending on their role in CME, might show regularity patterns other than intensity profiles, such as protein displacement.

To identify such patterns, our pipeline is split into two parts: 1) In a first data preprocessing and examination step, multiple characteristics are quantified for each track (movement speed, time of 50% intensity, time of maximum intensity, etc.). This first step makes it possible to search for and find patterns and regularities among the recorded tracks. In step 2, alignments are calculated based on the new observed patterns from step 1. As a result, conclusions can be drawn from an aligned and averaged set of tracks. In this study, two characteristics proved to be most useful for our alignments: inflection point (see below) and 50% raw intensity.

*Inflection point:* Quantified coat protein tracks can usually be divided into two phases: An initial, non-motile phase (a) during which protein accumulates on the plasma membrane with little detectable movement and a second phase (b) in which the protein moves, likely reflecting the time when the plasma membrane deforms into an invagination ($b_1$), and when the freed endocytic vesicle pinches off and moves into the cytoplasm ($b_2$). We used the transition point between these two phases (a, $b_1$), which we refer to as the inflection point $t_{inflection}$, for CME track alignment. To calculate $t_{inflection}$, first, the frame by frame displacement $d_t$ was calculated for all subsequent timepoints t

and t+1 in a track: $d_{t+1} = sqrt[(x_{t+1}+1-x_t)^2 + (y_{t+1}-y_t)^2]$. The signature behavior is that $d_t$ is flat in phase (a) and then suddenly shows a slope change upon the transition to phase ($b_1$). To identify $t_{inflection}$, $d_t$ is first linearized using linear regression $d_{t,linear} = linregress(d_t)$ and then $d_{t,linear}$ is subtracted from $d_t$. $t_{inflection}$ now is the minimum in the resulting function: $t_{inflection} = argmin(d_t - d_{t,linear})$.

*50% raw intensity:* The time point for 50% raw fluorescence intensity $t_{0.5,raw}$ can be helpful to align the tracks for endocytic proteins with short lifetimes and/or complex displacement patterns. This value is found at $Intensity(t_{0.5,raw})=0.5 * max(Intensity(t))$.

*Implementation:* The pipeline is implemented as a python software library with a Jupyter based user interface, which allows user-friendly and interactive access to its functionality. The pipeline is open source under a Berkeley Software Distribution (BSD) 3-clause license and can be found on Github: https://github.com/DrubinBarnes/YeastTrackAnalysis (*Schöneberg, 2019*; copy archived at https://github.com/elifesciences-publications/YeastTrackAnalysis).

## 3D-STORM imaging and analysis

GFP nanobody Alexa Fluorphore 647 conjugation: A 100 µl GFP nanobody solution at 71 µM (gt-250, ChromoTek) was conjugated to the Alexa Flurophore 647 NHS ester (A20006, Thermo Fisher Scientific) following standard procedures (*Mund et al., 2014*). After purification as described by *Mund et al. (2014)* we obtained an AF647 GFP-nanobody solution with a concentration of 49.3 µM and a labeling ratio of approximately 1.2 dye per nanobody.

Fission yeast fixation and AF647 GFP-nanobody labeling: fission yeast cells were fixed and labeled following a method commonly used for good actin structure preservation (*Kaplan and Ewers, 2015*). Low speed centrifugation (1000 rpm) was used to pellet yeast cells. Paraformaldehyde was directly added to the culture to a 4% final concentration, followed by 15 min incubation. Cells were washed twice with CS (cytoskeleton) buffer (10 mM MES, 150 mM NaCl, 5 mM EGTA, 5 mM Glucose, 5 mM $MgCl_2$, 0.005% $NaN_3$, pH6.1) containing 50 mM $NH_4Cl$, for 5 min with gentle shaking. The cells were collected and resuspended with CS buffer contain 5% Bovine serum albumin (BSA) and 0.25%(vol/vol) Triton X-100. After 30 min gentle shaking, AF647 GFP-nanobody was added to the fixed cells to a final concentration of 0.5 µM. The fixed cells were incubated with AF647 GFP-nanobody with gentle shaking overnight at 4°C.

Alexa Fluor 647-labeled cell samples were mounted on gel pads with STORM imaging buffer consisting of 5% (w/v) glucose, 100 mM cysteamine, 0.8 mg/mL glucose oxidase, and 40 µg/mL catalase in 1M Tris-HCl (pH 7.5) (*Huang et al., 2008*; *Rust et al., 2006*). Coverslips were sealed using Cytoseal 60. STORM imaging was performed on a homebuilt setup based on a modified Nikon Eclipse Ti-U inverted fluorescence microscope using a Nikon CFI Plan Apo λ 100x oil immersion objective (NA 1.45). Dye molecules were photoswitched to the dark state and imaged using a 647 nm laser (MPB Communications, Quebec, Canada); this laser beam passed through an acousto-optic tunable filter and an optical fiber into the back focal plane of the microscope and onto the sample at intensities of ~2 kW $cm^{-2}$. A translation stage was used to shift the laser beams towards the edge of the objective so light reached the sample at incident angles slightly smaller than the critical angle of the glass-water interface. A 405 nm laser was used concurrently with the 647 nm laser to reactivate fluorophores into the emitting state. The power of the 405 nm laser was adjusted (typical range 0–1 W $cm^{-2}$) during image acquisition so that at any given instant, only a small, optically resolvable fraction of the fluorophores in the sample were in the emitting state. For 3D STORM imaging, a cylindrical lens was inserted into the imaging path so that images of single molecules were elongated in opposite directions for molecules on the proximal and distal sides of the focal plane (*Huang et al., 2008*; *Rust et al., 2006*). The raw STORM data was analyzed according to previously described methods (*Huang et al., 2008*; *Rust et al., 2006*). Data were collected at a frame rate of 110 Hz, for a total of ~60,000 frames per image.

Endocytic sites were first identified through signals in the 488 nm channel. Conventional images of yeast containing puncta or bright spots were then overlaid with the image in the super-resolution channel by adding the super-resolution image to the conventional image. If signals in both the conventional and super-resolution channel overlapped, the structure in the super-resolution channel was cropped. The corresponding molecule list (includes X, Y, Z positions of all molecules in the reconstructed image) were generated from the cropped structures for further data analysis. If no signal overlap was observed between the conventional and super-resolution channel, the site was not

subjected to analysis. The puncta in crowded regions of the cell could not be clearly resolved as a single endocytic site and therefore were not included in our analysis.

The single molecule positions for the structures of interest obtained from the reconstructed STORM image were loaded into MATLAB. A rotation matrix was applied to the molecule position list such that the invagination site was rotated to a horizontal orientation. The length of the invagination was measured by computing the distance between every pair of molecules within a set threshold (~0.01 difference in pixel size) along the y-axis, until the longest distance was found. The distance between the two molecules was calculated by the distance formula ($d = \sqrt{(x_2-x_1)^2 + (y_2-y_2)^2}$). The length, in pixel size, was then converted to nanometers using the conversion scale one pixel = 160 nm. The position of the plasma membrane relative to the endocytic structures was estimated by the background signals showing the cell shape.

## Acknowledgements

We thank members of the Drubin/Barnes laboratory for helpful discussions. We are grateful to Paula Real Calderon in Fred Chang's laboratory for advice on fission yeast molecular biology.

We thank Emily Teranishi for help generating yeast strains. We also thank Dr. Samuel J Kenney for help in establishing the 3D-STORM experimental setup. We greatly appreciate Dr. Andrea Picco for his advice on particle tracking analysis. We also thank Dr. Jonathan Wong and Dr. Meiyan Jin for critically reading our manuscript. This work was supported by NIH grant R35GM118149 to DGD. Dr. Tom Pollard was a Miller Visiting Professor at UC Berkeley. This work was also supported by Pew Biomedical Scholars Award to KX.

## Additional information

### Funding

| Funder | Grant reference number | Author |
| --- | --- | --- |
| National Institutes of Health | R35GM118149 | David G Drubin |
| Pew Charitable Trusts | Pew Biomedical Scholars Award | Ke Xu |

The funders had no role in study design, data collection and interpretation, or the decision to submit the work for publication.

### Author contributions

Yidi Sun, Conceptualization, Data curation, Formal analysis, Supervision, Validation, Investigation, Visualization, Methodology, Writing—original draft, Writing—review and editing; Johannes Schöneberg, Software, Methodology, Writing—review and editing; Xuyan Chen, Data curation, Software, Formal analysis; Tommy Jiang, Data curation; Charlotte Kaplan, Formal analysis; Ke Xu, Resources, Supervision, Methodology; Thomas D Pollard, Conceptualization, Writing—review and editing; David G Drubin, Conceptualization, Writing—original draft, Writing—review and editing

### Author ORCIDs

Yidi Sun (iD) https://orcid.org/0000-0002-2157-1983
Ke Xu (iD) https://orcid.org/0000-0002-2788-194X
Thomas D Pollard (iD) https://orcid.org/0000-0002-1785-2969
David G Drubin (iD) https://orcid.org/0000-0003-3002-6271

### Decision letter and Author response

Decision letter https://doi.org/10.7554/eLife.50749.sa1
Author response https://doi.org/10.7554/eLife.50749.sa2

## Additional files

### Supplementary files
• Transparent reporting form

### Data availability
All data generated or analysed during this study are included in the manuscript and supporting files. Code for custom data analysis pipeline can be found at https://github.com/DrubinBarnes/Yeast-TrackAnalysis; copy archived at https://github.com/elifesciences-publications/YeastTrackAnalysis.

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
