## [Decision Letter]

**Decision letter after peer review:**

Thank you for submitting your article "Direct comparison of clathrin-mediated endocytosis in budding and fission yeast reveals conserved and evolvable features" for consideration by *eLife*. Your article has been reviewed by Suzanne Pfeffer as the Senior Editor, a Reviewing Editor, and three reviewers. The following individuals involved in review of your submission have agreed to reveal their identity: Vladimir Sirotkin (Reviewer #2).

The reviewers have discussed the reviews with one another and the Reviewing Editor has drafted this decision to help you prepare a revised submission. *eLife*

In the present manuscript, the authors employed several creative and rigorous approaches to systematically compare protein abundances, morphology and kinetics of late, actin-dependent steps of endocytosis in budding versus fission yeast. A number of previous studies have comparatively examined endocytosis in fission and budding yeast and reported significant differences in the copy numbers of several proteins at the endocytic structures. New data obtained in the present study suggest that protein abundances and molecular mechanisms of actin-mediated membrane invagination and scission are more similar in these two yeasts than previously thought. In particular, new experiments did not support previously proposed hypothesis of the essential role of two-zone actin polymerization in fission yeast. At the same time, several differences in protein concentrations and in dynamics of the endocytic process, for example, in the extent and rate of membrane invagination were found. Overall, while this study does not seem to offer a significant conceptual advance in the mechanistic understanding of endocytosis in yeast cells, the completion of this rigorous analysis is important for reconciling previous conflicting reports and models, thus creating much needed basis for future mechanistic studies. For this reason, it should be revised as an *eLife* RESOURCE.

1) With regard to presenting the value of this work to a wide reader of *eLife* who are not experts in yeast endocytosis, it is unclear whether observed differences have any functional implications. In other words, it is unclear if the abundances of proteins and the length/rate of membrane invagination substantially affect the rate and capacity of endocytosis. Are the internalization rates of cargo, such as transporters or receptors, different in fission and budding yeast? Or would observed differences provide opportunities for species-specific regulatory mechanisms? If not addressed by direct measurements, the authors must discuss these points and make the present manuscript more widely appreciated by the broad *eLife* readership.

2) Substantial homo-FRET likely occurs within the nanocage between closely positioned GFP molecules which could result in a reduced fluorescence signal per GFP molecule as compared to single-molecule measurements. Although this will not affect measurements of the relative concentrations of GFP-tagged proteins at the endocytic structures in two yeast types, their absolute protein copy numbers might be underestimated.

3) Do protein concentrations at the endocytic bud correlate with their total expression levels? In Figure 1C, total cellular concentration of spWsp1 appears to be higher in fission yeast than that of scLas17 in budding yeast, which correlates with their concentrations at endocytic structures. On the other hand, in Figure 1A, Pan1 is more abundant at endocytic structures in fission than in budding yeast but total Pan1 fluorescence intensities are similar in two yeasts. It would strengthen the manuscript if the information about total cellular levels of proteins listed in Supplementary file 1 in two yeasts is presented and discussed, if available from mass-spectrometry databases or microscopy studies.

4) The authors state in subsection “Absolute numbers of proteins at endocytic sites quantified using ratiometric comparison to fluorescence intensity of a 120-sfGFP-tagged nanocage” that 'the numbers we determined for several key fission yeast proteins were 1.5-2 times lower than previously reported (Figure 2E, 2F and 2G (Sirotkin et al., 2010)'. But actually, the difference is not that dramatic as one can see from calculating the ratios of numbers from Supplementary file 1: two Arp2/3 complex subunits, 1.05; WASp, 1.67; verprolin, 1.47; myosin-1, 2.35; Fim1, 1.35; HIP1R, 1.29; intersectin, 1.19. So, for 6 out of 8 proteins the ratio is less than 1.5, which was very satisfying to see. And while the WASp count might indeed be inflated in Sirotkin et al., (2010) due to high cytoplasmic background of Wsp1 and, as the current study suggests, background correction method used, greater than 2-fold difference for Myo1 is to me genuinely puzzling. In my hands, mGFP-Myo1 patches are always 1.5-2.0-fold brighter than mGFP-Wsp1 patches. While I do not think that reconciling this discrepancy is strictly necessary for the current study, it might be important to do it in the long-term. But I am wondering if the authors attempted to reconcile this discrepancy by (i) changing location of the tag to the N-terminus, (ii) imaging the mGFP-Myo1 strain from Sirotkin et al., (2010) side-by-side with their Myo1-GFP strain on their calibrated microscope, and (iii) sequencing their Myo1-GFP strain to make sure there are no mutations in the GFP?

5) Both N-terminally (Sirotkin et al., 2005, 2010) and C-terminally GFP-tagged Wsp1 proteins are only partially functional based on their synthetic lethality with myo1 deletion. Since the current study is going to be the definitive gold-standard description of timing of endocytic events in relation to the numbers of molecules, it is very important to provide a control for any possible effects of spGFP-Wsp1 on the dynamics of Fim1-mCherry. This can be easily done by imaging side-by-side GFP-Wsp1+Fim1-mCherry strain versus Fim1-mCherry alone strain.

6) One of the major conclusions from the work is that the endocytic invaginations in *S. pombe* seem to be longer and grow faster. On one hand, the longer length of the *S. pombe* endocytic invaginations could already be expected from comparing the live cell imaging and super-resoultion microscopy data on chemically fixed samples (Arasada et al., 2018; Mund et al., 2018), and from comparing electron microscopy images of endocytic invaginations in *S. cerevisiae* and *S. pombe* (Encinar del Dedo et al., 2014; Idrissi et al., 2008). The observation that the invaginations might grow faster is more interesting and novel, but the time of vesicle fission is only inferred in the manuscript by Sun and coworkers. Defining the dynamics of the scission machinery would be helpful in this context.

7) An interesting observation is that deletion of the CA domain of the *S. pombe* myosin-I does not influence the endocytic coat and actin dynamics, whereas deletion of that of N-WASP does. This is used to imply that the hypothesis proposing the existence of two distal actin networks organized by the myosin-I and N-WASP at the base and the tip of the *S. pombe* invaginations is not functionally relevant. Much care should be taken with this conclusion though. It might just be that the CA domains of the endocytic NPFs are redundant, as it has been shown for *S. cerevisiae*, only that in *S. pombe*, because the lower number of myosin-I molecules, N-WASP might be the prominent NPF. In this context, the authors are requested to combine the mutations of both NPFs. On the other hand, the CA domains of the endocytic NPFs might basically work to trigger the formation of a branched actin network on the endocytic coat at the beginning of the budding process, and this network might then be organized by the different actin binding proteins and even grow in the absence of further Arp2/3 complex activation. Please discuss here that the presence of two distinct actin networks associated with endocytic invaginations has also been proposed in *S. cerevisiae*, being the Arp2/3 complex associated with the network positioned at the tip, but not the base of the invagination (Idrissi et al., 2012; Idrissi et al., 2008). It should be mentioned that description of endocytic actin by in vivo imaging or super-resolution mostly uses Abp1 and Arp2/3 components as markers, and one should keep in mind that those will label mainly branched actin networks but do not necessarily follow all actin filaments at endocytic sites.

8) Missing information in the Materials and methods section:

(i) There is no strain table, which will be important for people wishing to request strains from this study.

(ii) There is no detailed description of how the genes were tagged. This is particularly important because the authors specifically test the effect of tag placement on molecular counts and dynamics of Las17.

(iii) Were tagged strains verified by sequencing? This is particularly important since a mutation in GFP tag (if present) can have an effect on fluorescence intensity and thereby skew the measurements.

(iv) Figure 1—figure supplement 2 shows that for that figure cells were attached to ConA-coated coverslips but no information is given how this was done.

---

## [Author Response]

1) With regard to presenting the value of this work to a wide reader of ELife who are not experts in yeast endocytosis, it is unclear whether observed differences have any functional implications. In other words, it is unclear if the abundances of proteins and the length/rate of membrane invagination substantially affect the rate and capacity of endocytosis.

This manuscript is now being designated as an *eLife* Resource, which we feel is appropriate because we provide a large amount of high quality data, which will be valuable to many researchers for years to come, and a robust new standard for counting molecules observed in fluorescence data, but have not yet ventured deeply into mechanism, though we speculate on possible mechanistic implications.

Are the internalization rates of cargo, such as transporters or receptors, different in fission and budding yeast?

At this point, information on rates of receptor-mediated cargo endocytosis are lacking in fission yeast. Because this manuscript is designated as a Resource, we feel that such an analysis is beyond the scope of the study.

Or would observed differences provide opportunities for species-specific regulatory mechanisms?

Thank you for the suggestion to compare regulation of endocytosis in the two yeasts. In subsection “Comparison of protein dynamics and endocytic site morphology for CME internalization stages” we added text suggesting that different turgor pressures in the two evolutionarily distant yeasts might account in part for the differences in the sizes of endocytic vesicles and the numbers of endocytic proteins. Turgor pressure might have contributed to selection of the differences during the divergent evolution of the two yeasts. Our data lay a solid foundation of investigating the mechanisms in the future. Such work may help to explain why clathrin-coated pits differ in size in various species (McMahon and Boucrot, 2011)

If not addressed by direct measurements, the authors must discuss these points and make the present manuscript more widely appreciated by the broad eLife readership.

Thank you for encouraging us to discuss in more detail how comparative studies contribute to understanding the mechanism of endocytosis. The high conservation of the proteins that drive endocytosis means that the common ancestor of fungi and animals had genes for these proteins, so the studies performed here and in other labs on yeast provide a foundation for understanding of CME in animals. Of course, all organisms that diverged from this common ancestor now provide examples of how proteins that arose from the ancient genes have been adapted to many lifestyles. To relate our findings to those in other eukaryotes, we list the names of the mammalian homologues in tables and figures (e.g., Table 1, Figure 1, Figure 2, etc).

Our results summarized in Figure 9 emphasize that yeasts have provided more details about molecular recruitment and function in CME pathways than in any other cell type. For example, fully understanding endocytic internalization depends on knowing local numbers of each participating protein and the corresponding membrane morphology. Acquisition of such information would be very difficult if not impossible to achieve in mammalian cells. Even with new genome editing methods and sophisticated new imaging modalities, it will be a long while before a complete set of mammalian cell lines with endogenously GFP-tagged endocytic proteins is available to measure systematically the numbers of proteins and their dynamics. Before this study, we did not know if the reported differences in protein abundances and membrane invagination dynamics in budding and fission yeast were meaningful. Therefore, our rigorous study not only contributes essential new knowledge about endocytosis in the two yeast species, but also provides the most detailed and precise information available for mathematical modeling studies to provide rigorous tests of mechanistic hypotheses. More importantly, our results are useful for future identification of conserved, essential principles (what is adaptable and what is not) of the endocytosis in general. We modified the Abstract, Introduction and Discussion section to emphasize these points.

In addition, this work reports technical advances in protein counting and particle tracking analysis that will benefit cell biology research in general.

2) Substantial homo-FRET likely occurs within the nanocage between closely positioned GFP molecules which could result in a reduced fluorescence signal per GFP molecule as compared to single-molecule measurements. Although this will not affect measurements of the relative concentrations of GFP-tagged proteins at the endocytic structures in two yeast types, their absolute protein copy numbers might be underestimated.

We see no evidence of homo-FRET. First, we verified our approach using a previously used kinetochore calibration (Figure 2D). Second, we have seen no evidence for this phenomenon when comparing 1-GFP with 3-GFP tags in budding yeast (Sun et al., 2006, Supplemental figure 2C Ark1-GFP vs Ark1-3GFP) and in fission yeast (Wu et al., 2005).

3) Do protein concentrations at the endocytic bud correlate with their total expression levels? In Figure 1C, total cellular concentration of spWsp1 appears to be higher in fission yeast than that of scLas17 in budding yeast, which correlates with their concentrations at endocytic structures. On the other hand, in Figure 1A, Pan1 is more abundant at endocytic structures in fission than in budding yeast but total Pan1 fluorescence intensities are similar in two yeasts. It would strengthen the manuscript if the information about total cellular levels of proteins listed in supplemental table 1 in two yeasts is presented and discussed, if available from mass-spectrometry databases or microscopy studies.

We appreciate this suggestion. First, to clarify, we do not measure concentrations at endocytic sites, but rather numbers of proteins at endocytic sites.

In response to the questions, we generated a new Table 1 with ratios of the peak number of molecules per patch to total molecules per cell for budding yeast. The total molecule numbers were based on the measurements from Ho et al. (Ho et al., 2018).

We also generated a Table 1 with ratios of peak molecules per patch to total molecules per cell for fission yeast. The numbers of total molecules were based on the measurement from Pollard lab (Wu et al., 2005 and Sirotkin et al., 2010).

These tables show that the number of proteins per patch do not scale with the concentrations of the free protein in the cytoplasm. This is expected, since the equilibrium binding constants surely vary. The binding rate constants may also be limiting in some cases, so those reactions will not come to equilibrium.

4) The authors state in subsection “Absolute numbers of proteins at endocytic sites quantified using ratiometric comparison to fluorescence intensity of a 120-sfGFP-tagged nanocage” that 'the numbers we determined for several key fission yeast proteins were 1.5-2 times lower than previously reported (Figure 2E, 2F and 2G (Sirotkin et al., 2010)'. But actually, the difference is not that dramatic as one can see from calculating the ratios of numbers from Supplementary file 1: two Arp2/3 complex subunits, 1.05; WASp, 1.67; verprolin, 1.47; myosin-1, 2.35; Fim1, 1.35; HIP1R, 1.29; intersectin, 1.19. So, for 6 out of 8 proteins the ratio is less than 1.5, which was very satisfying to see.

We agree with these observations, so we revised the section about published protein numbers in fission yeast with a more precise description (subsection “Absolute numbers of proteins at endocytic sites quantified using ratiometric comparison to fluorescence intensity of a 120-sfGFP-tagged nanocage”).

And while the WASp count might indeed be inflated in Sirotkin et al. (2010) due to high cytoplasmic background of Wsp1 and, as the current study suggests, background correction method used, greater than 2-fold difference for Myo1 is to me genuinely puzzling. In my hands, mGFP-Myo1 patches are always 1.5-2.0-fold brighter than mGFP-Wsp1 patches. While I do not think that reconciling this discrepancy is strictly necessary for the current study, it might be important to do it in the long-term. But I am wondering if the authors attempted to reconcile this discrepancy by (i) changing location of the tag to the N-terminus, (ii) imaging the mGFP-Myo1 strain from Sirotkin et al., (2010) side-by-side with their Myo1-GFP strain on their calibrated microscope, and (iii) sequencing their Myo1-GFP strain to make sure there are no mutations in the GFP?

To specifically address this reviewer’s concerns about the protein numbers for Myo1 relative to Wsp1, we directly compared the maximum intensity of mEGFP-Myo1 and mEGFP-Wsp1 at endocytic sites in the same field. The peak intensity of mEGFP-Myo1 is about 1.25 times higher than mEGFP-wsp1 (Author response image 1), confirming our protein counts in Figure 2F.

5) Both N-terminally (Sirotkin et al., 2005, 2010) and C-terminally GFP-tagged Wsp1 proteins are only partially functional based on their synthetic lethality with myo1 deletion. Since the current study is going to be the definitive gold-standard description of timing of endocytic events in relation to the numbers of molecules, it is very important to provide a control for any possible effects of spGFP-Wsp1 on the dynamics of Fim1-mCherry. This can be easily done by imaging side-by-side GFP-Wsp1+Fim1-mCherry strain versus Fim1-mCherry alone strain.

As the reviewer suggested, we made a side-by-side comparison (see Author response image 2) of the dynamics of Fim1-mCherry in a *GFP-wsp1+ fim1-mCherry* strain versus a *fim1-mCherry* alone strain (A and B). The dynamics of Fim1-mCherry are indistinguishable in these two strains (C, D, E). We conclude that the GFP tag on Wsp1 does not compromise the dynamics of Fim1 during endocytosis. However, curiously, the intensity of Fim1-mCherry appears to be slightly (~1.2 times) higher in the *GFP-wsp1+fim1-mCherry* strain than in the *fim1-mCherry* alone strain (F).

**Author response image 2. respfig2:** 

6) One of the major conclusions from the work is that the endocytic invaginations in S. pombe seem to be longer and grow faster. On one hand, the longer length of the S. pombe endocytic invaginations could already be expected from comparing the live cell imaging and super-resoultion microscopy data on chemically fixed samples (Arasada et al., 2018; Mund et al., 2018), and from comparing electron microscopy images of endocytic invaginations in S. cerevisiae and S. pombe (Encinar del Dedo et al., 2014; Idrissi et al., 2008). The observation that the invaginations might grow faster is more interesting and novel, but the time of vesicle fission is only inferred in the manuscript by Sun and coworkers. Defining the dynamics of the scission machinery would be helpful in this context.

Although Arasada et al.’s live-cell PALM analysis indicated that the actin network moves on average 300 nm into the cell, that study did not address the ultrastructural organization of actin and whether a connection to the plasma membrane was maintained. Using 3D-STORM, we collected data for over 5 minutes for many endocytic sites to produce reconstructions and to gain the spatial resolution required to determine that Fim1 assembles into a structure with the longest length of approx. 350 nm, which likely outlines and extends beyond the membrane invagination.

We were not aware that immuno-EM had been done in fission yeast when we wrote the manuscript. We appreciated learning about this work and have now referred to Encinar del Dedo et al., 2014 at the appropriate parts of the text to strengthen our conclusions (Introduction; subsection “Comparison of protein dynamics and endocytic site morphology for CME internalization stages”).

In most previous studies, the dynamics of coat proteins (Sla1, Pan1, Sla2), scission proteins (Rvs167 and Rvs161) and actin (Abp1) were used as spatial and temporal landmarks to predict the membrane shape. These studies assumed that these proteins appear when and where they exert their functions during endocytic invagination. We took an alternative approach to align numerous tracks of an endocytic protein (regardless its function) and calculated the average and standard deviation (SD) of displacement over time with high temporal resolution. The spatial displacements of various endocytic proteins transitioned sharply from low SD to high SD in both single color or two-color live-cell imaging. We interpret the transition as vesicle scission. Our approach confirmed previous observations on budding yeast and showed that the longer endocytic invagination in fission yeast elongate faster than in budding yeast.

We followed the reviewers’ suggestion to look at the scission machinery in fission yeast by examining the dynamics of Hob1-GFP together with Fim1-mCherry, because sequence similarity suggests that spHob1 is the fission yeast homologue of scRvs167, which has been implicated in scission (Kaksonen et al., 2005; Kishimoto et al., 2011). New Figure 5C and 5D and Figure 5—figure supplement 2 show that spHob1-GFP appears slightly after Fim1 (Figure 5D, dotted line 1) and rapidly reaches its peak intensity (Figure 5D, dotted line 2). The spHob1-GFP intensity begins to drop just before spFim1 reaches its peak intensity (Figure 5D, dotted line 3). The dynamics of Hob1 are consistent with the idea that Hob1 facilitates vesicle scission, and moves away from the cell surface with the retracting endocytic tubule, as observed for Rvs167 in budding yeast (Kaksonen ea al., 2005, Picco et al., 2015).

7) An interesting observation is that deletion of the CA domain of the S. pombe myosin-I does not influence the endocytic coat and actin dynamics, whereas deletion of that of N-WASP does. This is used to imply that the hypothesis proposing the existence of two distal actin networks organized by the myosin-I and N-WASP at the base and the tip of the S. pombe invaginations is not functionally relevant. Much care should be taken with this conclusion though. It might just be that the CA domains of the endocytic NPFs are redundant, as it has been shown for S. cerevisiae, only that in S. pombe, because the lower number of myosin-I molecules, N-WASP might be the prominent NPF. In this context, the authors are requested to combine the mutations of both NPFs.

We added Figure 8D with quantification of Fim1-mCherry dynamics in Wsp1-CA∆ strains. Subsection “The relative importance of spWsp1 and spMyo1 NPF activity for fission yeast endocytosis” explains that Fim1-mCherry puncta still turn over at the cell cortex in the Wsp1-CA∆ strain, implying that Myo1 alone can facilitate actin polymerization. However, only 8% of Fim1-mCherry puncta move inward in the *Wsp1-CA∆* strain, indicating that endocytosis is severely compromised. Thus, these observations suggest that actin nucleation by Wsp1 CA domain has a non-redundant role in endocytic internalization in fission yeast. We expect that deleting the CA domains of both NPFs simply disables the process entirely and likely kills the cells, as in the double deletion mutations of the two proteins.

On the other hand, the CA domains of the endocytic NPFs might basically work to trigger the formation of a branched actin network on the endocytic coat at the beginning of the budding process, and this network might then be organized by the different actin binding proteins and even grow in the absence of further Arp2/3 complex activation. Please discuss here that the presence of two distinct actin networks associated with endocytic invaginations has also been proposed in S. cerevisiae, being the Arp2/3 complex associated with the network positioned at the tip, but not the base of the invagination (Idrissi et al., 2012; Idrissi et al., 2008). It should be mentioned that description of endocytic actin by in vivo imaging or super-resolution mostly uses Abp1 and Arp2/3 components as markers, and one should keep in mind that those will label mainly branched actin networks but do not necessarily follow all actin filaments at endocytic sites.

Previously, immunoelectron microscopy detected HA-tagged Myo5 at both the base and tip of the invaginated endocytic membrane in chemically fixed budding yeast cells (Idrissi et al., 2008; Idrissi et al., 2012). These observations led to speculation that two distinct endocytic actin networks exist in budding yeast. However, two independent live-cell imaging studies observed that Myo5-GFP remains at the cell surface during the lifetime of endocytic patches (Galletta et al., 2008; Sun et al., 2006). In addition, super-resolution microscopy suggested that Myo5-GFP forms a nano-template for actin nucleation at the membrane base (Mund et al., 2018). For these reasons it seems very likely that most if not all type I myosin is localized at the base of invaginated endocytic membranes. In response to the reviewer’s comment, we now discuss these points (Figure 1—figure supplement 1 legend).

In budding yeast, the dynamic behavior of Abp1and Arp2/3 complex are identical to Sac6 (Fimbrin, a marker for filamentous actin) (Kaksonen et al., 2005 and Martin et al., 2006), providing confidence that collectively these proteins act as faithful actin reporters.

8) Missing information in the Experimental Procedures:(i) There is no strain table, which will be important for people wishing to request strains from this study.(ii) There is no detailed description of how the genes were tagged. This is particularly important because the authors specifically test the effect of tag placement on molecular counts and dynamics of Las17.(iii) Were tagged strains verified by sequencing? This is particularly important since a mutation in GFP tag (if present) can have an effect on fluorescence intensity and thereby skew the measurements.

We thank the reviewer for identifying this omission and added a strain table as Table 1. Most of the strains weused for molecular counting were used/published in previously studies (Kaksonen et al., 2005; Sun et al., 2006, Sirotkin et al., 2010). All constructs are amplified by high fidelity PCR and we have never detected altered GFP fluorescence. The Materials and methods section provides information on making chromosomally N-terminal tagged Las17. The *GFP* coding sequence followed by a gctgccgcagcggct linker was integrated into the chromosome before the *LAS17* coding region by Gibson assembly cloning (NEB, Inc) and yeast transformation. The resulting strain was verified by DNA sequencing.

(iv) Figure 1—figure supplement 2 shows that for that figure cells were attached to ConA-coated coverslips but no information is given how this was done.

We added this information to the Materials and methods section. For live budding yeast cell imaging, cells were grown to early log phase at 25ºC. The cells in synthetic media were adhered to the surface of a concanavalin A coated coverslip, which was then inverted onto a glass slide, and sealed with vacuum grease (Dow Corning).